# Rethinking Joint Maximum Mean Discrepancy for Domain Adaptation

**Wei Wang**[1]    **Haifeng Xia**[1]    **Chao Huang**[1*]    **Zhengming Ding**[2]
**Cong Wang**[3]    **Haojie Li**[4]    **Xiaochun Cao**[1]
[1]Shenzhen Campus of Sun Yat-sen University  [2]Department of Computer Science, Tulane University
[3]University of California, San Francisco  [4]Shandong University of Science and Technology
{wangwei29, xiahf5, huangch253, caoxiaochun}@mail.sysu.edu.cn   zding1@tulane.edu
supercong94@gmail.com   hjli@sdust.edu.cn

## Abstract

In domain adaption (DA), joint maximum mean discrepancy (JMMD), as a famous distribution-distance metric, aims to measure joint probability distribution difference between the source domain and target domain, while it is still not fully explored and especially hard to be applied into a subspace-learning framework as its empirical estimation involves a tensor-product operator whose partial derivative is difficult to obtain. To solve this issue, we deduce a concise JMMD based on the Representer theorem that avoids the tensor-product operator and obtains two essential findings. First, we reveal the uniformity of JMMD by proving that previous marginal, class conditional, and weighted class conditional probability distribution distances are three special cases of JMMD with different label reproducing kernels. Second, inspired by graph embedding, we observe that the similarity weights, which strengthen the intra-class compactness in the graph of Hilbert Schmidt independence criterion (HSIC), take opposite signs in the graph of JMMD, revealing why JMMD degrades the feature discrimination. This motivates us to propose a novel loss JMMD-HSIC by jointly considering JMMD and HSIC to promote discrimination of JMMD. Extensive experiments on several cross-domain datasets could demonstrate the validity of our revealed theoretical results and the effectiveness of our proposed JMMD-HSIC.

## 1 Introduction

Domain adaptation (DA) has emerged as an effective technology to solve the well-known problem of domain discrepancy that frequently occurs in reality [1–5]. Many promising approaches have been suggested to mitigate this issue from different perspectives [6–10]. A major issue in DA is how to formulate a favorable probability distribution distance that can be applied to measure the proximity of two different probability distributions, thus numerous probability distribution-distance metrics have been proposed over the years. For example, the Quadratic and Kullback-Leibler distances derived from the Bregman divergence and generated by different convex functions are introduced to match two different probability distributions explicitly [11]. However, extending them into different models may be inflexible since they are parametric and require an intermediate density estimation [12]. The Wasserstein distance derived from the optimal transport problem exploits a transportation plan to align two different marginal [13], class conditional [14] or joint [15] probability distributions, but it might be inconvenient to be applied into a subspace-learning framework because it often comes down to a complex bi-level optimization problem [16].

---

*Corresponding author.

39th Conference on Neural Information Processing Systems (NeurIPS 2025).

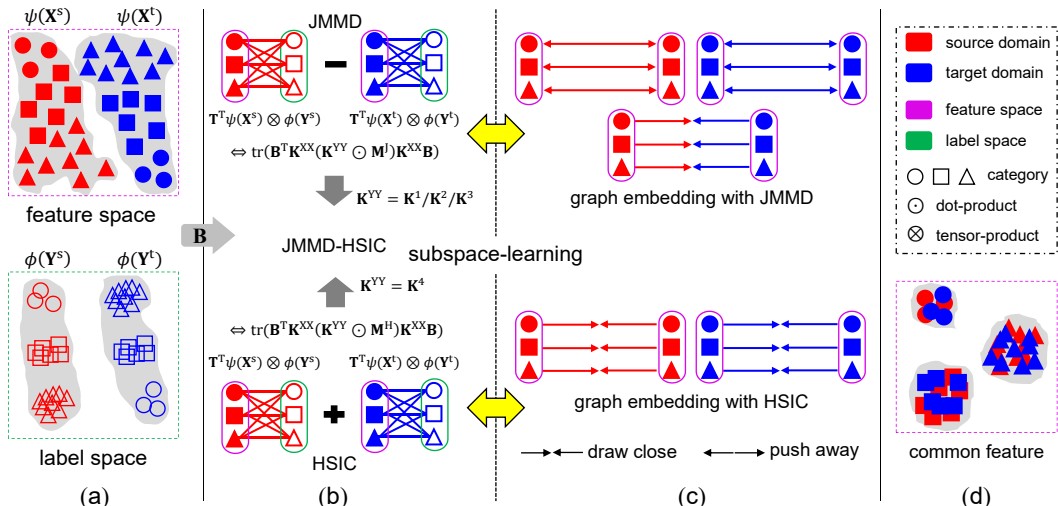

Figure 1: The overview of our revealed theoretical results and proposed JMMD-HSIC. (a) We map the features (upper part) and labels (lower part) of the source and target domains into the RKHS, respectively; (b) The application of JMMD (upper part) and HSIC (lower part) in a subspace-learning framework; (c) The graph embedding interpretation of JMMD (upper part) and HSIC (lower part) in a subspace-learning framework; (d) Learned feature representations of the source and target domains after subspace learning. '$-$' in the module of JMMD means the feature-label dependence difference, and '$+$' in the module of HSIC means separately considering feature-label dependence in the two domains. $\psi$ and $\phi$ are feature and label mappings for a RKHS. $\mathbf{T}$ and $\mathbf{B}$ are projection matrices for a projected RKHS. $\mathbf{T}\psi(\mathbf{X}) \otimes \phi(\mathbf{Y})$ is a tensor-product operator for the covariance. $\text{tr}(\mathbf{B}^\top \mathbf{K}^{XX}(\mathbf{K}^{YY} \odot \mathbf{M}^{J/H})\mathbf{K}^{XX}\mathbf{B})$ is the deduced concise JMMD/HSIC in a projected RKHS.

Gretton *et al.* propose a metric of maximum mean discrepancy (MMD), which empirically estimates the distance between two probability distributions in a reproducing kernel Hilbert space (RKHS) [17]. Due to its simplicity and solid theoretical foundation, it has been applied into a wide range of problems, such as deep generative models [18] and variational autoencoders [19], *etc*. Although MMD could establish marginal [12], class conditional [20], and weighted class conditional [21] probability distribution distances, it is still not fully explored about the joint probability distribution distance (*i.e.*, joint maximum mean discrepancy, JMMD) and is hard to be applied into a subspace-learning framework, as its empirical estimation involves a tensor-product operator whose partial derivative is difficult to obtain.

Specifically, when JMMD is considered in a subspace-learning framework, we project data features and their corresponding labels into a RKHS (Fig. 1(a)), and aim to exploit a projected RKHS by a feature projection matrix $\mathbf{T}$ with infinite dimensions (the upper part of Fig. 1(b)). In the projected RKHS or the common feature space, the feature-label dependence difference between the source domain and target domain is minimized, and their joint probability distributions [22] are aligned accordingly (Fig. 1(d)). Here, the covariance that involves a tensor-product operator describes the feature-label dependence. However, one issue remains to be overcome in this process: the partial derivative with respect to the infinite-dimensional $\mathbf{T}$ is hard to obtain. In this paper, based on the Representer theorem [23], we deduce a concise JMMD that avoids the tensor-product operator and optimizes a finite-dimensional $\mathbf{B}$ instead of $\mathbf{T}$, where the Representer theorem represents $\mathbf{T}^\top \psi(\mathbf{x})$ by finite values of a kernel function with corresponding coefficients from $\mathbf{B}$, and some matrix operation properties make the tensor-product disappear. Therefore, the derivative can be easily obtained.

With the concise JMMD, we obtain two essential findings. Firstly, we reveal the uniformity of JMMD by proving that previous popular marginal, class conditional, and weighted class conditional probability distribution distances are its three special cases with label kernels $\mathbf{K}^1$, $\mathbf{K}^2$ and $\mathbf{K}^3$, and we also prove that they are reproducing kernels. This finding could provide theoretical guidance to refine JMMD by designing label kernels for different problems in DA. Moreover, recent work indicates that the procedure of distribution alignment degrades the feature discrimination unexpectedly [24, 25]

but lacks theoretical support. To reveal this, similar to JMMD, we deduce a concise Hilbert Schmidt independence criterion (HSIC) [26], which models the feature-label dependence and maximizes the dependence to improve feature discrimination (*the features have better discrimination ability if the feature-label dependence is larger and vice versa*) (the lower part of Fig. 1(b)). We design a label reproducing kernel $\mathbf{K}^4$ for HSIC to strengthen the intra-class compactness or feature discrimination. Then, we explore the relationship between JMMD and HSIC inspired by the graph embedding viewpoint to better understand the reason for feature discrimination degradation in JMMD.

Specifically, we observe that the similarity weights, which strengthen intra-class compactness in the graph of HSIC, take opposite signs in the graph of JMMD. Thus, JMMD degrades the discrimination unexpectedly. As shown in Fig. 1(c), in the feature space, JMMD tries to *push data points from the same classes in the same domain further* and draw those from the same classes in different domains closer. In contrast, in the feature space, HSIC aims to *draw data points from the same classes in the same domain closer* (intra-class compactness). To this end, we propose a novel loss JMMD-HSIC by jointly considering JMMD and HSIC. Therefore, we can improve the discrimination of JMMD, leading to a better DA capacity (Fig. 1(d)). The whole pipeline of this paper is briefly depicted in Fig. 1, and our main contributions are summarized below,

- Based on the Representer theorem, we deduce a concise JMMD that avoids the tensor-product operator, and the derivative can be easily obtained so that it can be applied into a subspace-learning framework.
- With the concise JMMD, we reveal its uniformity by proving that previous distribution distances are its special cases with different label reproducing kernels. This finding yields theoretical guidance for refining JMMD by designing label kernels for different problems.
- To better understand the reason for feature discrimination degradation in JMMD, we reveal the relationship between JMMD and HSIC inspired by graph embedding. Then, a novel loss JMMD-HSIC is proposed to promote discrimination of JMMD.

## 2 Related Work

### 2.1 Maximum Mean Discrepancy

Gretton *et al*. introduce a probability distribution-distance metric MMD [17]. Along this direction, Pan *et al*. incorporate MMD into a subspace-learning framework to learn some transfer components [12] across the source domain and target domain. Duan *et al*. embed MMD into a multiple kernel learning framework to jointly learn a kernel function and a robust classifier [27] by minimizing the structural risk functional and the distribution mismatch. Long *et al*. down-weighted source instances irrelevant to target ones to realize more positive knowledge transfer based on MMD [28]. Ghifary *et al*. employ MMD to deal with distribution bias in a simple neural network [29]. Tzeng *et al*. propose a domain confusion loss based on MMD in a deep neural network [30]. Long *et al*. present a deep adaptation network where the multi-kernel MMD is applied into all task-specific layers [31].

To model more complicated probability distribution distances, Long *et al*. propose a class-wise MMD to align class conditional probability distributions [20], which can be conveniently optimized in many subspace-learning frameworks [32, 33]. To deal with an imbalanced dataset, Wang *et al*. establish a weighted class-wise MMD where class prior probabilities are introduced [21]. Concerning the label distribution shift problem, Yan *et al*. construct a weighted MMD where source instances are multiplied by special weights [34]. Moreover, Wang *et al*. raise a dynamic balanced MMD to quantitatively account for relative importance between the marginal and class conditional probability distribution distances [35]. Deng *et al*. propose an extended MMD to simultaneously minimize the intra-class dispersion and maximize the inter-class compactness [36].

Zhang *et al*. estimate the joint probability distribution discrepancy based on the Bayesian law and propose a discriminative joint probability distribution discrepancy [37]. In contrast, Long *et al*. estimate the uncentered feature-label covariance in the source and target domains, and consider the difference in covariance between the two domains as the joint probability distribution discrepancy. Then, they propose JMMD and construct a transfer network accordingly [22]. However, JMMD is still not fully explored and is very hard to be applied to a subspace-learning framework as it involves a tensor-product operator whose partial derivative is difficult to obtain. This paper theoretically deduces a concise JMMD and obtains two essential findings.

## 2.2 Hilbert Schmidt Independence Criterion

HSIC measures dependence between two random variables [26]. To preserve important data features, Dorri *et al*. regard original and transformed data features as two random variables and maximize their dependence during distribution alignment [38, 39]. Yan *et al*. minimize dependence between projected features and domain features (*e.g.*, background information) to learn a robust domain-invariant subspace [40]. Inspired by classifier adaptation, Wang *et al*. propose a projected HSIC to maximize feature-label dependence in a reconstruction DA framework [41]. Similarly, we aim to maximize the feature-label dependence while we further explore the relationship between JMMD and HSIC, and design a label reproducing kernel $\mathbf{K}^4$ for HSIC in this work. Besides, HSIC is computed on the whole domains [41] while we separately compute HSIC on the source and target domains.

## 2.3 Graph Embedding

Graph embedding as a prevalent technology has been applied to many problems due to its advantage in discovering complicated data structures and relationships, such as subspace-learning methods [42, 43] and graph convolutional networks [44, 45], *etc*. A lot of subspace-learning-based DA approaches aim to establish a similarity graph and exploit a feature mapping to embed the graph vertices into a desirable low-dimensional subspace so that some important data structures (local manifold structure [35], local discrimination structure [24], *etc*.) could be respected elegantly during the distribution alignment process. This paper does not focus on leveraging the graph embedding technique in the DA problem. Instead, we analyze the relationship between JMMD and HSIC inspired by graph embedding to illustrate the reason for feature discrimination degradation in JMMD.

# 3 Rethinking JMMD

In this paper, the bold uppercase letter $\mathbf{X}$ denotes a matrix, while the bold lowercase letter $\mathbf{x}$ is a column vector of $\mathbf{X}$. Moreover, $\mathbf{x}_i$ is the i-th column vector of $\mathbf{X}$ and $x_{ij}$ is the value of i-th row and j-th column of $\mathbf{X}$. $\otimes$ and $\odot$ are the tensor-product and dot-product operators.

**Domain Adaptation.** Given a labeled source domain $\mathcal{D}^s := (\mathbf{X}^s, \mathbf{Y}^s)$ and an unlabeled target domain $\mathcal{D}^t := (\mathbf{X}^t, \mathbf{Y}^t)$, where $\mathbf{X}^{s/t} \in \mathbb{R}^{m \times n^{s/t}}$, $\mathbf{Y}^{s/t} \in \mathbb{R}^{C \times n^{s/t}}$, 'm' is the feature dimension, '$n^{s/t}$' is the number of source/target instances ($n^s + n^t = n$) and 'C' is the number of shared categories. DA assumes that the two domains follow different joint probability distributions, *i.e.*, $\mathcal{P}^s(\mathbf{X}^s, \mathbf{Y}^s) \neq \mathcal{P}^t(\mathbf{X}^t, \mathbf{Y}^t)$, but share the same feature and label spaces, and $\mathbf{Y}^t$ is not available during the training process. $\mathbf{Y}^{s/t}$ is the probability soft label, *i.e.*, the probability of a data sample $\mathbf{x}_i$ that belongs to a class 'c' is $y_{ci}$. Notably, the hard label (one-hot label) is a special case of $\mathbf{Y}^{s/t}$ where $y_{ci} = 1$ and $y_{\bar{c}i} = 0$ ($\bar{c} \neq c$) if $\mathbf{x}_i$ belongs to the c-th class. DA aims to design a distribution-distance metric to minimize the divergence between their joint probability distributions so that the classifier trained on a source domain can be generalized into another target domain.

**Reproducing Kernel Hilbert Space.** RKHS is a Hilbert space ($\mathcal{H}$) of function $f : \Omega \rightarrow \mathbb{R}$ on a domain $\Omega$, and its inner-product and Hilbert-Schmidt norm are $\langle \cdot, \cdot \rangle_{\mathcal{H}}$ and $|| \cdot ||_{\mathcal{H}}$. The evaluation functional $f(\mathbf{x})$ can be reproduced by a reproducing kernel function $k(\mathbf{x}, \mathbf{x}^\top)$, *i.e.*, $\langle f(\cdot), k(\mathbf{x}, \cdot) \rangle_{\mathcal{H}} = f(\mathbf{x})$, and the RKHS takes its name from this so-called reproducing kernel function. $k(\mathbf{x}, \cdot)$ can be viewed as an implicit mapping $\psi(\mathbf{x})$ (infinite dimensions) where $k(\mathbf{x}, \mathbf{x}^\top) = \langle \psi(\mathbf{x}), \psi(\mathbf{x}^\top) \rangle_{\mathcal{H}}$.

## 3.1 A Concise JMMD

In DA, JMMD as a famous probability distribution-distance metric is still not fully explored and hard to be applied into a subspace-learning framework as its empirical estimation involves a tensor-product operator whose partial derivative is nontrivial to obtain [22]. In this section, we first deduce a concise JMMD and reveal its uniformity. Then, we deduce a concise HSIC to explore the relationship between JMMD and HSIC inspired by graph embedding, revealing why JMMD degrades the feature discrimination. Finally, we propose a novel loss JMMD-HSIC to improve the discrimination of JMMD and apply it into a general subspace-learning framework.

Given a domain $\mathcal{D}$ sampled from a joint probability distribution $\mathcal{P}(\mathbf{X}, \mathbf{Y})$ [46], we project the data feature $\mathbf{x}$ and its corresponding label $\mathbf{y}$ into a RKHS, *i.e.*, $\psi(\mathbf{x})$ and $\phi(\mathbf{y})$, where $\psi$ and $\phi$ are the

feature and label mappings. Then, we utilize the uncentered covariance between the feature and label to represent a joint probability distribution, *i.e.*, $\mathcal{C}_{XY} := \mathbb{E}_{XY}(\psi(\mathbf{x}) \otimes \phi(\mathbf{y}))$. Here, we estimate the covariance $\mathcal{C}_{XY}$ of domain $\mathcal{D}$ with the expectation $\mathbb{E}_{XY}$ or mean $\mu_{XY}$ of all samples' covariance $\psi(\mathbf{x}) \otimes \phi(\mathbf{y})$ in a RKHS. The covariance essentially describes the dependence between feature and label. Due to the impossibility of obtaining all possible samples in domain $\mathcal{D}$ (an infinite number), JMMD [22] adopts the maximum likelihood estimate principle and utilizes finite samples to empirically estimate $\mu_{X^sY^s}$ and $\mu_{X^tY^t}$ for source domain and target domain, and then minimizes the loss of $||\mu_{X^sY^s} - \mu_{X^tY^t}||_{\mathcal{H}}^2$ to draw joint probability distributions of the two domains closer. Formally, JMMD and its concise form in a RKHS is defined as below,

$$\mathbb{D}_{\mathcal{H}}(\mathcal{P}^s(\mathbf{X}^s, \mathbf{Y}^s), \mathcal{P}^t(\mathbf{X}^t, \mathbf{Y}^t)) = ||\mu_{X^sY^s} - \mu_{X^tY^t}||_{\mathcal{H}}^2 = \text{tr}(\mathbf{K}^{XX}(\mathbf{K}^{YY} \odot \mathbf{M}^J)), \tag{1}$$

where $\mathbf{K}^{XX}, \mathbf{K}^{YY} \in \mathbb{R}^{n \times n}$ are the kernel matrices and they are computed by $k_{ij}^{XX} = k^X(\mathbf{x}_i, \mathbf{x}_j^\top)$, $k_{ij}^{YY} = k^Y(\mathbf{y}_i, \mathbf{y}_j^\top)$. $k^X$ and $k^Y$ are feature and label kernels. $\mathbf{M}^J \in \mathbb{R}^{n \times n}$ is the MMD matrix for JMMD. Remarkably, the nonlinear functions $\psi$ and $\phi$ do not need to be explicit, and the tensor-product operator disappears, more details could be found in ***Section A of the supplementary material***. To incorporate JMMD into a subspace-learning framework, we deduce the concise form of JMMD in a projected RKHS as below,

$$||\frac{1}{n^s}\sum_{i=1}^{n^s}(\mathbf{T}^\top\psi(\mathbf{x}_i^s) \otimes \phi(\mathbf{y}_i^s)) - \frac{1}{n^t}\sum_{j=1}^{n^t}(\mathbf{T}^\top\psi(\mathbf{x}_j^t) \otimes \phi(\mathbf{y}_j^t))||_{\mathcal{H}}^2 = \text{tr}(\mathbf{B}^\top\mathbf{K}^{XX}(\mathbf{K}^{YY} \odot \mathbf{M}^J)\mathbf{K}^{XX}\mathbf{B}). \tag{2}$$

As shown in the left side of Eq. (2), the dimension of feature projection matrix $\mathbf{T} \in \mathbb{R}^{\infty \times d}$ is infinite since $\psi$ is an infinite mapping, and a tensor-product operator is involved so that it is nontrivial to obtain the partial derivative with respect to $\mathbf{T}$. To overcome this issue, we utilize the Representer theorem and some matrix operation properties to obtain the right side of Eq. (2). Remarkably, the infinite-dimensional $\mathbf{T}$ does not need to be optimized because we resort to optimize a finite-dimensional $\mathbf{B}$, and the tensor-product operator disappears. Therefore, it is easy to obtain the partial derivative with respect to $\mathbf{B}$, and JMMD could be applied into a subspace-learning framework. More details about the proof of Eq. (2) can be found in ***Section B of the supplementary material***.

### 3.2 The Uniformity of JMMD

In this section, we introduce a theorem to illustrate that JMMD is a unified form of existing popular marginal, class conditional, and weighted class conditional probability distribution distances.

**Theorem 1** *The marginal, class conditional, and weighted class conditional probability distribution distances are three special cases of JMMD with label reproducing kernels $K^1$, $K^2$ and $K^3$, and more details about these three distances could be found in **Section C of the supplementary material**. $K^1 = \mathbf{1}_{n \times n}$ is a matrix whose elements are all 1 with size of $n \times n$, and $K^2$, $K^3$ are defined as below,*

$$k_{ij}^2 = \begin{cases} (n^sn^s)/(n^{s,c}n^{s,c}), & \mathbf{x}_i \in \mathcal{D}^{s,c}, \mathbf{x}_j \in \mathcal{D}^{s,c} \\ (n^tn^t)/(n^{t,c}n^{t,c}), & \mathbf{x}_i \in \mathcal{D}^{t,c}, \mathbf{x}_j \in \mathcal{D}^{t,c} \\ (n^sn^t)/(n^{s,c}n^{t,c}), & \mathbf{x}_i \in \mathcal{D}^{s,c}, \mathbf{x}_j \in \mathcal{D}^{t,c} \\ (n^tn^s)/(n^{t,c}n^{s,c}), & \mathbf{x}_i \in \mathcal{D}^{t,c}, \mathbf{x}_j \in \mathcal{D}^{s,c} \\ 0, & \text{otherwise,} \end{cases} \quad k_{ij}^3 = \begin{cases} 1, & \mathbf{x}_i \in D^{s,c}, \mathbf{x}_j \in \mathcal{D}^{s,c} \\ 1, & \mathbf{x}_i \in D^{t,c}, \mathbf{x}_j \in \mathcal{D}^{t,c} \\ 1, & \mathbf{x}_i \in D^{s,c}, \mathbf{x}_j \in \mathcal{D}^{t,c} \\ 1, & \mathbf{x}_i \in D^{t,c}, \mathbf{x}_j \in \mathcal{D}^{s,c} \\ 0, & \text{otherwise,} \end{cases} \tag{3}$$

where the superscript 's/t,c' denotes data points from the c-th class in the source/target domain. The proof of this theorem could be found in ***Section D of the supplementary material***, and we also prove that $\mathbf{K}^1$, $\mathbf{K}^2$ and $\mathbf{K}^3$ are the reproducing kernels in ***Section E of the supplementary material***. Notably, Theorem 1 yields theoretical guidance to refine JMMD by designing more delicate label kernels for different problems in DA, and we will leave this open direction in our future work.

### 3.3 A Concise HSIC

HSIC [26] also utilizes the covariance to establish the feature-label dependence, and aims to maximize the dependence for a given domain to improve its feature discrimination. A problem is that the domain-specific dependence may be decreased or the feature discrimination is degraded unexpectedly when minimizing JMMD. In the next section, we will illustrate the reason for discrimination degradation in JMMD from the graph embedding viewpoint. Following JMMD, we utilize finite samples to empirically estimate $\mathcal{C}_{X^s Y^s}$ and $\mathcal{C}_{X^t Y^t}$, and maximize these two terms separately. Similarly, HSIC involves a tensor-product operator whose derivative is hard to obtain, thus we deduct a concise HSIC in an RKHS and a concise HSIC in a projected RKHS as follows,

$$\text{tr}(\mathbf{K}^{XX}(\mathbf{K}^{YY} \odot \mathbf{M}^{H})), \quad \text{tr}(\mathbf{B}^{\top}\mathbf{K}^{XX}(\mathbf{K}^{YY} \odot \mathbf{M}^{H})\mathbf{K}^{XX}\mathbf{B}). \tag{4}$$

Notably, the concise HSIC is consistent with JMMD. Thus, it is easy to jointly consider them in a concise form, which will be introduced in 3.5. Besides, we design a label kernel for HSIC as below,

$$k_{ij}^4 = \begin{cases} (-n^s n^s)/(n^{s,c} n^{s,c}), & \mathbf{x}_i, \mathbf{x}_j \in \mathcal{D}^{s,c}, i \neq j \\[2mm] \left(n^s n^s (n^{s,c} - 1)\right)/\left(n^{s,c} n^{s,c}\right), & \mathbf{x}_i, \mathbf{x}_j \in \mathcal{D}^{s,c}, i{=}j \\[2mm] (-n^t n^t)/(n^{t,c} n^{t,c}), & \mathbf{x}_i, \mathbf{x}_j \in \mathcal{D}^{t,c}, i \neq j \\[2mm] \left(n^t n^t (n^{t,c} - 1)\right)/\left(n^{t,c} n^{t,c}\right), & \mathbf{x}_i, \mathbf{x}_j \in \mathcal{D}^{t,c}, i{=}j \\[2mm] 0, & \text{otherwise.} \end{cases} \tag{5}$$

There are some advantages with $\mathbf{K}^4$: 1) (4) will be a minimization problem which can be easily analyzed from the graph embedding viewpoint; 2) In 3.4, we will illustrate that the intra-class compactness (discrimination) of source domain and target domain can be improved with label kernel $\mathbf{K}^4$; 3) $\mathbf{K}^4$ is a reproducing kernel which is proved in ***Section E of the supplementary material***.

### 3.4 A Graph Embedding Viewpoint

In this section, we reveal that JMMD degrades the feature discrimination from the graph embedding viewpoint. Given a data matrix $\mathbf{X} = [\mathbf{x}_1, \cdots, \mathbf{x}_n]$, we establish a nearest neighbor graph $G$ with n vertices, where each vertex denotes a data point. Let $\mathbf{W}$ be the weight matrix of $G$, and $w_{ij}$ measures the similarity weight between $\mathbf{x}_i$ and $\mathbf{x}_j$ in original feature space (the larger $w_{ij}$ is and the closer they are and vice versa). Graph embedding technique [42] tries to find a desirable feature representation of $\mathbf{X}$ so that it could respect the relationship between each two data points in the original feature space. Formally, it aims to minimize the following objective function,

$$\sum_{i=1}^{n} \sum_{j=1}^{n} \left(w_{ij} \|\mathbf{z}_i - \mathbf{z}_j\|_2^2\right) = \text{tr}(\mathbf{Z}\mathbf{L}\mathbf{Z}^{\top}), \tag{6}$$

where $\mathbf{Z}$ is the embedding representation of $\mathbf{X}$. $\mathbf{L} = \mathbf{Q} - \mathbf{W}$ is the graph Laplacian matrix where $\mathbf{Q} = \mathbf{diag}(\mathbf{q}_1, \cdots, \mathbf{q}_n)$ is a diagonal matrix and $\mathbf{q}_i = \sum_{j=1}^{n} w_{ij}$. Inspired by graph embedding, we regard $\mathbf{L}^J = \mathbf{K}^2 \odot \mathbf{M}^J$ and $\mathbf{L}^H = \mathbf{K}^4 \odot \mathbf{M}^H$ as two graph Laplacian matrices. Then, we reveal the similarity weight matrices $\mathbf{W}^J$ and $\mathbf{W}^H$ of JMMD and HSIC as follows,

$$w_{ij}^J = \begin{cases} -1/(n^{s,c} n^{s,c}), & \mathbf{x}_i \in \mathcal{D}^{s,c}, \mathbf{x}_j \in \mathcal{D}^{s,c} \\[2mm] -1/(n^{t,c} n^{t,c}), & \mathbf{x}_i \in \mathcal{D}^{t,c}, \mathbf{x}_j \in \mathcal{D}^{t,c} \\[2mm] 1/(n^{s,c} n^{t,c}), & \mathbf{x}_i \in \mathcal{D}^{s,c}, \mathbf{x}_j \in \mathcal{D}^{t,c} \\[2mm] 1/(n^{t,c} n^{s,c}), & \mathbf{x}_i \in \mathcal{D}^{t,c}, \mathbf{x}_j \in \mathcal{D}^{s,c} \\[2mm] 0, & \text{otherwise,} \end{cases} \qquad w_{ij}^H = \begin{cases} 1/(n^{s,c} n^{s,c}), & \mathbf{x}_i \in \mathcal{D}^{s,c}, \mathbf{x}_j \in \mathcal{D}^{s,c} \\[2mm] 1/(n^{t,c} n^{t,c}), & \mathbf{x}_i \in \mathcal{D}^{t,c}, \mathbf{x}_j \in \mathcal{D}^{t,c} \\[2mm] 0, & \text{otherwise.} \end{cases}$$

$$\tag{7}$$

Then, the concise JMMD (2) and HSIC (4) in a projected RKHS can be rewritten as follows,

$$\sum_{i=1}^{n} \sum_{j=1}^{n} (w_{ij}^{J}||\mathbf{B}^{\top}\mathbf{k}_{i}^{XX} - \mathbf{B}^{\top}\mathbf{k}_{j}^{XX}||_{F}^{2}), \quad \sum_{i=1}^{n} \sum_{j=1}^{n} \left( w_{ij}^{H}||\mathbf{B}^{\top}\mathbf{k}_{i}^{XX} - \mathbf{B}^{\top}\mathbf{k}_{j}^{XX}||_{F}^{2} \right), \quad (8)$$

where $\mathbf{B}^{\top}\mathbf{k}^{XX}$ is the embedded feature representation and (8) is similar to (6). Specifically, the distance between $\mathbf{B}^{\top}\mathbf{k}_{i}^{XX}$ and $\mathbf{B}^{\top}\mathbf{k}_{j}^{XX}$ should be closer if $w_{ij} > 0$ but further if $w_{ij} < 0$ since the goal is to minimize (8). From (7), we observe that JMMD will *push two data points from the same classes in the same domain further*, and draw those from the same classes in different domains closer. HSIC will *draw two data points from the same classes in the same domain closer* (intra-class compactness). These observations illustrate that JMMD degrades feature discrimination as the similarity weights which strengthen intra-class compactness in the graph of HSIC, take opposite signs in the graph of JMMD. Besides, the designed $\mathbf{K}^{4}$ makes (4) a minimization problem.

## 3.5 The Proposed JMMD-HSIC

From the above analysis, we consider JMMD and HSIC by jointly minimizing (1)/(2) and (4) to propose JMMD-HSIC. With a little abuse of notations, we denote $\mathbf{K}^{J}$ and $\mathbf{K}^{H}$ for label kernels of JMMD and HSIC, then JMMD-HSIC in an RKHS and a projected RKHS are finalized as follows,

$$\text{tr}(\mathbf{K}^{XX}(\mathbf{K}^{J} \odot \mathbf{M}^{J} + \delta\mathbf{K}^{H} \odot \mathbf{M}^{H})), \quad \text{tr}(\mathbf{B}^{\top}\mathbf{K}^{XX}(\mathbf{K}^{J} \odot \mathbf{M}^{J} + \delta\mathbf{K}^{H} \odot \mathbf{M}^{H})\mathbf{K}^{XX}\mathbf{B}), \quad (9)$$

where $\delta$ aims to balance the relative importance between JMMD and HSIC, and we can adopt $\mathbf{K}^{1}$, $\mathbf{K}^{2}$ or $\mathbf{K}^{3}$ for $\mathbf{K}^{J}$ and $\mathbf{K}^{4}$ for $\mathbf{K}^{H}$. To validate our revealed theoretical results and the effectiveness of JMMD-HSIC, we incorporate it into a general subspace-learning framework, *i.e.*, $\arg\min \mathcal{L}^{\star}(\mathbf{B}) + \mathcal{L}(\mathbf{B})$, where $\mathcal{L}^{\star}(\mathbf{B})$ is the JMMD-HSIC loss in a projected RKHS, and its partial derivative with respect to $\mathbf{B}$ is $2\mathbf{K}^{XX}(\mathbf{K}^{J} \odot \mathbf{M}^{J} + \delta\mathbf{K}^{H} \odot \mathbf{M}^{H})\mathbf{K}^{XX}\mathbf{B}$. $\mathcal{L}(\mathbf{B})$ is a general form for other losses.

# 4 Experiments

## 4.1 Datasets and Experimental Settings

To validate our revealed theoretical results and the effectiveness of JMMD-HSIC, we conduct extensive experiments on four benchmark datasets in cross-domain object recognition. $D^{1}$: **Office10-Caltech10** [47] consists of four domains, *i.e.*, Amazon, Dslr, Webcam, Caltech; $D^{2}$: **ImageCLEF-DA** includes three domains, *i.e.*, Caltech-256, ImageNet ILSVRC 2012, Pascal VOC 2012; $D^{3}$: **Office-31** [48] contains three domains, *i.e.*, Amazon, Dslr, Webcam; $D^{4}$: **Office-Home** [49] has four domains, *i.e.*, Art, Clipart, Product, Real-world.

As this paper mainly focuses on the problem that JMMD is hard to be applied into a subspace-learning framework, we incorporate the proposed JMMD-HSIC into three subspace-learning-based DA approaches, *i.e.*, joint distribution adaptation (JDA) [20], selective pseudo-labeling (SPL) [50], and optimal graph learning-based label propagation (OGL$^{2}$P) [51]. We abbreviate these three variants as JDA+JMMD-HSIC, SPL+JMMD-HSIC, and OGL$^{2}$P+JMMD-HSIC, respectively. For a fair comparison, all hyper-parameters remain consistent with the three approaches. Regarding $\delta$, we uniformly set it to 0.5 for JDA+JMMD-HSIC, while assigning different values on the corresponding datasets for the other two variants after trials. On Office10-Caltech10, we use the SURF features with 800 dimensions [47] and the DECAF-6 features with 4096 dimensions [52]. On the other three datasets, we utilize the ResNet-50 features with 2048 dimensions [53]. Moreover, we adopt $\mathbf{K}^{2}$ for JMMD due to its superiority and $\mathbf{K}^{4}$ for HSIC.

## 4.2 Results

We compare our proposed approach with existing state-of-the-art shallow (SPL [50], PGCD [54], RMMD [55]) and deep DA approaches (BSP+MetaReg [56], DRDA [57], RSDA-MSTN [58], Jeffreys-DD [59], OGL$^{2}$P [51]) on $D^{3}$ and $D^{4}$. As can be seen from Tabs. 1 and 2, our proposed approach is better than the baseline methods SPL and OGL$^{2}$P on average, and has achieved 1.4%/0.8% and 0.9%/0.8% improvements on the two datasets, respectively. Besides, OGL$^{2}$P+JMMD-HSIC could

Table 1: Comparison results on Office-31 with ResNet-50 features. A, D, W in the second row denotes domains of Amazon, Dslr, and Webcam, respectively.

| Source | Venue | Amazon | | Dslr | | Webcam | | Avg. |
|---|---|---|---|---|---|---|---|---|
| Target | | D | W | A | W | A | D | |
| PGCD [54] | TIP'23 | 95.2 | 94.0 | 76.4 | 99.0 | 76.5 | 100.0 | 90.2 |
| RMMD-I [55] | TNNLS'23 | 90.4 | 88.4 | 74.1 | 98.7 | 74.8 | 99.8 | 87.7 |
| BSP+MetaReg [56] | TKDE'23 | 96.2 | 95.2 | 76.8 | 99.2 | 74.6 | 100.0 | 90.3 |
| DRDA [57] | TIP'23 | 94.5 | 95.8 | 75.6 | 98.8 | 76.6 | 100.0 | 90.2 |
| RSDA-MSTN [58] | TPAMI'24 | 96.1 | 95.9 | 77.8 | 99.3 | 78.2 | 100.0 | 91.2 |
| Jeffreys-DD [59] | NeurIPS'24 | 95.9 | 94.9 | 76.0 | 99.1 | 74.6 | 100.0 | 90.1 |
| SPL [50] | AAAI'20 | 93.0 | 92.7 | 76.4 | 98.7 | 76.8 | 99.8 | 89.6 |
| SPL+JMMD-HSIC | - | 95.8 | 95.5 | 78.5 | 99.1 | 77.0 | 100.0 | 91.0 |
| OGL$^2$P [51] | TIP'25 | 96.2 | 95.5 | 77.5 | 98.7 | 76.8 | 99.4 | 90.7 |
| OGL$^2$P+JMMD-HSIC | - | 96.5 | 95.8 | 78.8 | 99.3 | 78.8 | 100.0 | **91.5** |

Table 2: Comparison results on Office-Home with ResNet-50 features. A, C, P, R in the second row denotes domains of Artistic, Clipart, Product, and Real-World, respectively.

| Source | Venue | Artistic | | | Clipart | | | Product | | | Real-World | | | Avg. |
|---|---|---|---|---|---|---|---|---|---|---|---|---|---|---|
| Target | | C | P | R | A | P | R | A | C | R | A | C | P | |
| PGCD [54] | TIP'23 | 57.7 | 77.2 | 79.1 | 59.1 | 74.3 | 72.7 | 61.2 | 54.2 | 79.3 | 70.0 | 58.4 | 82.7 | 68.8 |
| RMMD-I [55] | TNNLS'23 | 58.4 | 77.8 | 79.3 | 61.6 | 72.8 | 73.0 | 62.7 | 55.3 | 78.9 | 70.4 | 60.1 | 83.2 | 69.5 |
| BSP+MetaReg [56] | TKDE'23 | 58.0 | 75.5 | 78.9 | 65.0 | 74.7 | 75.0 | 67.9 | 57.2 | 81.8 | 74.7 | 63.5 | 83.8 | 71.3 |
| DRDA [57] | TIP'23 | 58.2 | 74.2 | 81.2 | 65.6 | 75.1 | 73.3 | 65.8 | 57.1 | 80.4 | 75.6 | 63.2 | 85.1 | 71.2 |
| RSDA-MSTN [58] | TPAMI'24 | 59.6 | 79.2 | 81.1 | 68.7 | 77.7 | 77.7 | 67.8 | 61.0 | 82.2 | 75.3 | 60.8 | 85.9 | 73.1 |
| Jeffreys-DD [59] | NeurIPS'24 | 55.5 | 74.9 | 79.5 | 64.3 | 73.8 | 73.9 | 63.9 | 54.7 | 81.3 | 75.2 | 61.6 | 84.2 | 70.2 |
| SPL [50] | AAAI'20 | 54.5 | 77.8 | 81.9 | 65.1 | 78.0 | 81.1 | 66.0 | 53.1 | 82.8 | 69.9 | 55.3 | 86.0 | 71.0 |
| SPL+JMMD-HSIC | - | 56.8 | 77.1 | 81.6 | 66.5 | 79.4 | 81.2 | 67.9 | 55.0 | 83.4 | 70.9 | 57.1 | 85.8 | 71.9 |
| OGL$^2$P [51] | TIP'25 | 57.8 | 78.8 | 82.1 | 68.4 | 81.6 | 80.4 | 68.9 | 56.6 | 82.9 | 71.7 | 59.1 | 85.0 | 72.8 |
| OGL$^2$P+JMMD-HSIC | - | 58.3 | 79.6 | 82.5 | 69.3 | 81.9 | 80.9 | 69.5 | 57.9 | 83.3 | 73.4 | 61.2 | 85.3 | **73.6** |

achieve the best average results among all compared approaches, which has achieved 0.3% and 0.5% improvements compared with the second-best methods, *i.e.*, RSDA-MSTN. The comparison results on the other datasets could be found in ***Section F of the supplementary material***. Generally speaking, these results can show the effectiveness and competitiveness of our proposed JMMD-HSIC.

## 4.3 Feature Visualization

To further show the results of JMMD-HSIC, we visualize the feature distributions using the t-SNE algorithm as a common practice in this field [31, 22]. Fig. 2 shows the related results for JMMD, HSIC, and JMMD-HSIC in the SPL framework. The better the matching of points with the same color but different shapes, the smaller the distribution difference; disregarding shape, the tighter the clustering of points with the same color, the better the discriminability. As illustrated in Fig. 2(a), the original features perform badly on both the distribution alignment and discrimination. From Fig. 2(b), JMMD tries to align the feature distributions of the source domain and target domain, but damages the discrimination greatly. As depicted in Fig. 2(c), HSIC aims to enhance the discriminative structure in both the source and target domains. However, it exhibits poor distribution alignment, as highlighted by the dashed boxes where points of the same color but different shapes are distributed in different regions. As shown in Fig. 2(d), the feature distributions are matched better compared to HSIC (as highlighted by the dashed boxes where points of the same color but different shapes are distributed in the same region) and data points from the same classes tend to be closer (better discrimination) compared to JMMD. These observations could validate our revealed theoretical results and the effectiveness of JMMD-HSIC compared to JMMD and HSIC.

Table 3: Ablation study with different losses on Office-31 (average accuracy on 6 tasks) and Office-Home (average accuracy on 12 tasks) with ResNet-50 features.

| Dataset | SPL | SPL | | | OGL$^2$P | OGL$^2$P | | |
|---|---|---|---|---|---|---|---|---|
| | | JMMD | HSIC | JMMD-HSIC | | JMMD | HSIC | JMMD-HSIC |
| Office-31 | 89.6 | 87.3 | 89.9 | **91.0** | 90.7 | 88.5 | 90.6 | **91.5** |
| Office-Home | 71.0 | 68.7 | 69.0 | **71.9** | 72.8 | 71.3 | 71.5 | **73.6** |

Table 4: Ablation study with different label kernels on Office-31 (average accuracy on 6 tasks) and Office-Home (average accuracy on 12 tasks) with ResNet-50 features.

| Dataset | SPL | SPL+JMMD-HSIC | | | OGL$^2$P | OGL$^2$P+JMMD-HSIC | | |
|---|---|---|---|---|---|---|---|---|
| | | $\mathbf{K}^1$ | $\mathbf{K}^2$ | $\mathbf{K}^3$ | | $\mathbf{K}^1$ | $\mathbf{K}^2$ | $\mathbf{K}^3$ |
| Office-31 | 89.6 | 85.6 | **91.0** | 90.9 | 90.7 | 86.7 | **91.5** | 91.5 |
| Office-Home | 71.0 | 64.9 | **71.9** | 71.8 | 72.8 | 69.8 | **73.6** | 73.6 |

## 4.4 Ablation Study

We further validate our revealed theoretical results by inspecting the feature distribution distance (JMMD metric) and feature-label independence. The smaller the JMMD metric, the better the distribution alignment; the smaller the feature-label independence, the better the discriminability. We run the methods of PCA, JDA+JMMD, JDA+HSIC and JDA+JMMD-HSIC on the dataset of $D^1$ with SURF features and utilize four different classifiers (1-nearest neighbor (1-NN), SVM, label propagation (LP) [60] and nearest class prototype (NCP) [50]) and two different label forms (hard and soft). We report average results of the two metrics on all DA tasks. Then, we compute these two metrics of each method on their embedded feature representations. Note that, in order to compute the true distance or metric, we have to use the ground-truth labels instead of the pseudo ones. However, the ground-truth target labels are only used for verification, not for training procedure [20]. As shown in Fig. 3, we could obtain the following observations. With JMMD, the JMMD metric is the smallest among the four methods but the feature-label independence is the largest among them, which indicates good distribution alignment but compromises discriminative structure. Conversely, with HSIC, the feature-label independence is the smallest among the four methods but the JMMD metric is the largest among them, which suggests consideration of discriminative structure but overlooking distribution alignment. In contrast, the proposed JMMD-HSIC strikes a good balance between distribution alignment and discriminability, which could lead to a better DA capacity. Moreover, PCA performs poorly in both aspects. In Tab. 3, we conduct ablation experiments using different loss functions on baselines of SPL and OGL$^2$P and have the following observations. Considering that the classifiers employed in SPL and OGL$^2$P frameworks require higher feature discriminability, we can observe that HSIC outperforms JMMD. Since the Office-31 dataset has smaller distribution discrepancies, the performance improvement of HSIC over JMMD is more significant on the Office-31 dataset (2.6%, 2.1%) compared to the Office-Home dataset (0.3%, 0.2%). Our proposed loss function achieves optimal results. Although both SPL and OGL$^2$P also incorporate loss functions for feature distribution alignment and discriminative feature learning, our approach provides more precise balancing between these two objectives by fundamentally analyzing how JMMD compromises feature discriminability. In other words, we more effectively mitigate the impact of distribution alignment on feature discriminability. Consequently, our method achieves superior performance compared to SPL and OGL$^2$P. These observations could validate our revealed theoretical results and the effectiveness of proposed JMMD-HSIC compared to JMMD and HSIC.

In Tab. 4, we conduct ablation experiments to validate why we adopt the label kernel $\mathbf{K}^2$. It can be observed that the marginal distribution kernel $\mathbf{K}^1$, which neglects label information, leads to inaccurate distribution alignment and thus performs the worst. The results of the weighted class-conditional distribution kernel $\mathbf{K}^3$ and the conditional distribution kernel $\mathbf{K}^2$ are nearly identical because the class imbalance issue in the experimental dataset used in this paper is not severe—whereas the weighted class-conditional distribution kernel is specifically designed to address class imbalance.

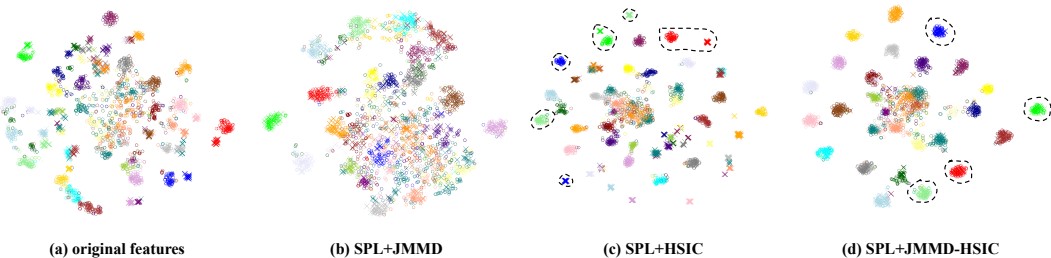

(a) original features   (b) SPL+JMMD   (c) SPL+HSIC   (d) SPL+JMMD-HSIC

Figure 2: Feature visualization of the DA task Amazon (source domain) → Webcam (target domain) from Office-31 dataset for SPL+JMMD, SPL+HSIC, and SPL+JMMD-HSIC. Different colors represent various classes, and '○' and '×' represent source domain and target domain, respectively.

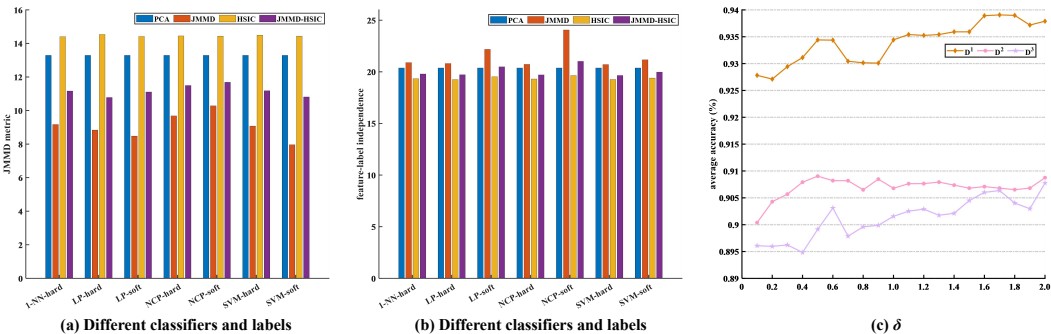

(a) Different classifiers and labels   (b) Different classifiers and labels   (c) $\delta$

Figure 3: Quantitative analysis for the JMMD (distribution distance, (a)) and HSIC (feature-label independence, (b)) metrics and sensitivity analysis for $\delta$ (c).

## 4.5 Sensitivity of Hyper-Parameter

We conduct the sensitivity analysis of $\delta$ in SPL+JMMD-HSIC to validate that the optimal results could be achieved under a stable range. We report average accuracy results of SPL+JMMD-HSIC on $D^1$, $D^2$, and $D^3$ with deep features. We plot average classification accuracy *w.r.t.*, different values of $\delta$ in Fig. 3(c), and choose $\delta \in [0.1, 2]$, which reflects the importance of HSIC for discrimination reinforcement in JMMD. It can be observed that the average accuracy often achieves its optimal value on a wide range for each dataset, which could display the stability of $\delta$. Notably, the average results of $D^4$ with different values of $\delta$ fluctuate between 68.1%∼71.7%, smaller than those of $D^1 \sim D^3$. Therefore, we do not plot the change for $D^4$ to observe the fluctuation trends more distinctively. The rationale behind selecting $\delta$ within the range of [0.1, 2] is as follows: we are gradually mitigating the negative effects of JMMD on discriminability when $\delta \in [0.1, 1]$. We completely eliminate the negative effects of JMMD, and gradually promote discriminability when $\delta \in [1, 2]$. Therefore, the model performs better when $\delta$ takes larger values within this range.

## 5 Conclusions

JMMD is still not fully explored and is especially hard to be applied to a subspace-learning framework. To overcome this problem, we deduce a concise JMMD and obtain two essential findings, *i.e.*, the uniformity of JMMD and the reason for feature discrimination degradation in JMMD. To strengthen the discrimination of JMMD, we jointly consider JMMD and HSIC to propose a novel loss dubbed as JMMD-HSIC. Comprehensive tests carried out on some benchmark datasets could validate our revealed theoretical results, and show promising performance with our proposed JMMD-HSIC. For future work, we plan to explore: **1**) designing an advanced label kernel to handle more diverse domain adaptation scenarios; **2**) extending the framework to more complex visual tasks; **3**) designing novel label kernels by systematically analyzing how different classifiers vary in their sensitivity to feature distribution alignment and feature discriminability.

## Acknowledgments

This work was supported in part by the the National Key R&D Program of China (Grant No.2022ZD0119200), National Natural Science Foundation of China (Grant No.62306343, No.62301621, No.62025604), Guangdong Basic and Applied Basic Research Foundation (Grant No.2025A1515011322, No.2025A1515011398), China Postdoctoral Science Foundation (Grant No.2025T180435, No.2024M753741). Shenzhen Science and Technology Program (No.20231121172359002, 2023A008), Shenzhen General Research Project (No.JCYJ20241202125904007).

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

# A  A Concise JMMD in an RKHS

**Theorem 2** *In an RKHS, JMMD could be rewritten as the following concise form,*

$$\mathbb{D}_{\mathcal{H}}\left(\mathcal{P}^s(\mathbf{X}^s, \mathbf{Y}^s), \mathcal{P}^t(\mathbf{X}^t, \mathbf{Y}^t)\right) = \left\|\left| \frac{1}{n^s}\sum_{i=1}^{n^s}\left(\psi(\mathbf{x}_i^s) \otimes \phi(\mathbf{y}_i^s)\right) - \frac{1}{n^t}\sum_{j=1}^{n^t}\left(\psi(\mathbf{x}_j^t) \otimes \phi(\mathbf{y}_j^t)\right) \right|\right\|_{\mathcal{H}}^2$$

$$= \mathrm{tr}\left(\mathbf{K}^{XX}(\mathbf{K}^{YY} \odot \mathbf{M}^J)\right),$$

(10)

where $\mathbb{D}_{\mathcal{H}}$ denotes a distance metric between two joint probability distributions, *i.e.*, $\mathcal{P}^s$ and $\mathcal{P}^t$. We empirically estimate JMMD with the following steps: i) we utilize $\psi$ and $\phi$ to map features and labels from the source domain and target domain to the RKHS, respectively; ii) we calculate the mean of the tensor product between feature and label for each domain; iii) we compute the difference between these two means. Notably, $\mathbf{x}_i$ and $\mathbf{x}_j$ are the i-th and j-th column vectors of $\mathbf{X}$.

Proof:

For convenience, we define $\Gamma^s(\mathbf{x}^s, \mathbf{y}^s)$ and $\Gamma^t(\mathbf{x}^t, \mathbf{y}^t)$ as shown in the following equations,

$$\Gamma^s(\mathbf{x}^s, \mathbf{y}^s) = \left[\psi(\mathbf{x}_1^s) \otimes \phi(\mathbf{y}_1^s), \cdots, \psi(\mathbf{x}_{n^s}^s) \otimes \phi(\mathbf{y}_{n^s}^s)\right] \in \mathbb{R}^{\infty \times n^s},$$

$$\Gamma^t(\mathbf{x}^t, \mathbf{y}^t) = \left[\psi(\mathbf{x}_1^t) \otimes \phi(\mathbf{y}_1^t), \cdots, \phi(\mathbf{x}_{n^t}^t) \otimes \phi(\mathbf{y}_{n^t}^t)\right] \in \mathbb{R}^{\infty \times n^t}.$$

(11)

Then, (10) could be rewritten as below,

$$\mathbb{D}_{\mathcal{H}}\left(\mathcal{P}^s(\mathbf{X}^s, \mathbf{Y}^s), \mathcal{P}^t(\mathbf{X}^t, \mathbf{Y}^t)\right) = \left\|\left| \frac{1}{n^s}\sum_{i=1}^{n^s}\left(\psi(\mathbf{x}_i^s) \otimes \phi(\mathbf{y}_i^s)\right) - \frac{1}{n^t}\sum_{j=1}^{n^t}\left[\psi(\mathbf{x}_j^t) \otimes \phi(\mathbf{y}_j^t)\right] \right|\right\|_{\mathcal{H}}^2$$

$$= \mathrm{tr}\left(\begin{bmatrix} \Gamma^s(\mathbf{x}^s, \mathbf{y}^s)^\top \Gamma^s(\mathbf{x}^s, \mathbf{y}^s) & \Gamma^s(\mathbf{x}^s, \mathbf{y}^s)^\top \Gamma^t(\mathbf{x}^t, \mathbf{y}^t) \\ \Gamma^t(\mathbf{x}^t, \mathbf{y}^t)^\top \Gamma^s(\mathbf{x}^s, \mathbf{y}^s) & \Gamma^t(\mathbf{x}^t, \mathbf{y}^t)^\top \Gamma^t(\mathbf{x}^t, \mathbf{y}^t) \end{bmatrix} \begin{bmatrix} \frac{\mathbf{1}_{n^s \times 1}\mathbf{1}_{n^s \times 1}^\top}{n^s n^s} & \frac{-\mathbf{1}_{n^s \times 1}\mathbf{1}_{n^t \times 1}^\top}{n^s n^t} \\ \frac{-\mathbf{1}_{n^t \times 1}\mathbf{1}_{n^s \times 1}^\top}{n^t n^s} & \frac{\mathbf{1}_{n^t \times 1}\mathbf{1}_{n^t \times 1}^\top}{n^t n^t} \end{bmatrix}\right).$$

(12)

where $\mathbf{1}_{n^s \times 1}$ and $\mathbf{1}_{n^t \times 1}$ are two column vectors whose elements are all ones with sizes of $n^s$ and $n^t$.

Moreover, we have the following equations,

$$\Gamma^s(\mathbf{x}^s, \mathbf{y}^s)^\top \Gamma^s(\mathbf{x}^s, \mathbf{y}^s) = \mathbf{K}^{X^s X^s} \odot \mathbf{K}^{Y^s Y^s},$$

(13)

$$\Gamma^t(\mathbf{x}^t, \mathbf{y}^t)^\top \Gamma^t(\mathbf{x}^t, \mathbf{y}^t) = \mathbf{K}^{X^t X^t} \odot \mathbf{K}^{Y^t Y^t},$$

(14)

$$\Gamma^s(\mathbf{x}^s, \mathbf{y}^s)^\top \Gamma^t(\mathbf{x}^t, \mathbf{y}^t) = \mathbf{K}^{X^s X^t} \odot \mathbf{K}^{Y^s Y^t},$$

(15)

$$\Gamma^t(\mathbf{x}^t, \mathbf{y}^t)^\top \Gamma^s(\mathbf{x}^s, \mathbf{y}^s) = \mathbf{K}^{X^t X^s} \odot \mathbf{K}^{Y^t Y^s}.$$

(16)

where $\mathbf{K}^{X^s X^s}, \cdots, \mathbf{K}^{Y^s Y^s}, \cdots \in \mathbb{R}^{n^s \times n^s}, \cdots \mathbb{R}^{n^s \times n^s}, \cdots$ are the kernel matrices and they are computed by $k_{ij}^{X^s X^s} = k^X(\mathbf{x}_i^s, \mathbf{x}_j^{s\top}), \cdots, k_{ij}^{Y^s Y^s} = k^Y(\mathbf{y}_i^s, \mathbf{y}_j^{s\top}), \cdots$. Here $k^X$ and $k^Y$ are feature and label kernels.

Therefore, we can rewrite (12) using the feature kernel matrix $\mathbf{K}^{XX}$ and the label kernel matrix $\mathbf{K}^{YY}$ as below,

$$\mathbb{D}_{\mathcal{H}}\Big(\mathcal{P}^s(\mathbf{X}^s, \mathbf{Y}^s), \mathcal{P}^t(\mathbf{X}^t, \mathbf{Y}^t)\Big)$$

$$= \text{tr}\left(\begin{bmatrix} \mathbf{K}^{X^sX^s} & \mathbf{K}^{X^sX^t} \\ \mathbf{K}^{X^tX^s} & \mathbf{K}^{X^tX^t} \end{bmatrix} \odot \begin{bmatrix} \mathbf{K}^{Y^sY^s} & \mathbf{K}^{Y^sY^t} \\ \mathbf{K}^{Y^tY^s} & \mathbf{K}^{Y^tY^t} \end{bmatrix} \begin{bmatrix} \frac{\mathbf{1}_{n^s \times 1}\mathbf{1}_{n^s \times 1}^\top}{n^s n^s} & \frac{-\mathbf{1}_{n^s \times 1}\mathbf{1}_{n^t \times 1}^\top}{n^s n^t} \\ \frac{-\mathbf{1}_{n^t \times 1}\mathbf{1}_{n^s \times 1}^\top}{n^t n^s} & \frac{\mathbf{1}_{n^t \times 1}\mathbf{1}_{n^t \times 1}^\top}{n^t n^t} \end{bmatrix}\right) \quad (17)$$

$$= \text{tr}\big(\mathbf{K}^{XX} \odot \mathbf{K}^{YY}\mathbf{M}^J\big),$$

where $\mathbf{K}^{XX} \in \mathbb{R}^{n \times n}$ and $\mathbf{K}^{YY} \in \mathbb{R}^{n \times n}$ are feature and label kernel matrices for all source and target domains, and $n = n^s + n^t$.

According to $\text{tr}\big(\mathbf{A} \odot \mathbf{BC}\big) = \text{tr}\big(\mathbf{A}(\mathbf{B} \odot \mathbf{C})\big)$ where $\mathbf{A}, \mathbf{B}$ and $\mathbf{C}$ are symmetric matrices [61], thus $\text{tr}\big(\mathbf{K}^{XX} \odot \mathbf{K}^{YY}\mathbf{M}^J\big) = \text{tr}\big(\mathbf{K}^{XX}(\mathbf{K}^{YY} \odot \mathbf{M}^J)\big)$. $\mathbf{M}^J$ is calculated as below,

$$m_{ij}^J = \begin{cases} 1/(n^s n^s), & \mathbf{x}_i, \mathbf{x}_j \in \mathcal{D}^s \\ 1/(n^t n^t), & \mathbf{x}_i, \mathbf{x}_j \in \mathcal{D}^t \\ -1/(n^s n^t), & \text{otherwise.} \end{cases} \quad (18)$$

where $\mathcal{D}^s$ and $\mathcal{D}^t$ denote source and target domains.

$\square$

# B  A Concise JMMD in a Projected RKHS

**Theorem 3** *In a projected RKHS, JMMD could be rewritten as the following concise form,*

$$\mathbb{D}_{\mathcal{H}}\Big(\mathcal{P}^s(\mathbf{X}^s, \mathbf{Y}^s), \mathcal{P}^t(\mathbf{X}^t, \mathbf{Y}^t)\Big)$$

$$= \left\| \frac{1}{n^s} \sum_{i=1}^{n^s} \Big(\mathbf{T}^\top \psi(\mathbf{x}_i^s) \otimes \phi(\mathbf{y}_i^s)\Big) - \frac{1}{n^t} \sum_{j=1}^{n^t} \Big(\mathbf{T}^\top \psi(\mathbf{x}_j^t) \otimes \phi(\mathbf{y}_j^t)\Big) \right\|_{\mathcal{H}}^2$$

$$= \left\| \frac{1}{n^s} \sum_{i=1}^{n^s} \Big(\sum_{l=1}^{n} \big(\mathbf{b}_l \psi(\mathbf{x}_l)^\top \psi(\mathbf{x}_i^s)\big) \otimes \phi(\mathbf{y}_i^s)\Big) - \frac{1}{n^t} \sum_{j=1}^{n^t} \Big(\sum_{l=1}^{n} \big(\mathbf{b}_l \psi(\mathbf{x}_l)^\top \psi(\mathbf{x}_j^t)\big) \otimes \phi(\mathbf{y}_j^t)\Big) \right\|_{\mathcal{H}}^2$$

$$= \text{tr}\Big(\mathbf{B}^\top \mathbf{K}^{XX}(\mathbf{K}^{YY} \odot \mathbf{M}^J)\mathbf{K}^{XX}\mathbf{B}\Big).$$

$$(19)$$

where $\mathbf{T} \in \mathbb{R}^{\infty \times d}$ is the feature projection matrix and 'd' is the dimension in the embedded subspace. Different from (10), we project $\psi(\mathbf{x}_i^s)$ and $\psi(\mathbf{x}_j^t)$ into an embedded subspace, and then empirically estimate JMMD.

Proof:

We begin by introducing the Representer theorem [23] as below,

**Theorem 4 (Representer theorem)** *It says that any function can be decomposed into finite values of a kernel function with corresponding coefficients [23].*

$$\mathbf{T}^\top \psi(\mathbf{x}) = \sum_{i=1}^{n} \Big(\mathbf{b}_i k^X(\mathbf{x}, \mathbf{x}_i)\Big) =$$

$$\sum_{i=1}^{n} \Big(\mathbf{b}_i \langle \psi(\mathbf{x}), \psi(\mathbf{x}_i) \rangle\Big) = \sum_{i=1}^{n} \Big(\mathbf{b}_i \psi(\mathbf{x}_i)^\top \psi(\mathbf{x})\Big),$$

$$(20)$$

where $\mathbf{b}_i \in \mathbb{R}^{d \times 1}$ and we definite a new projection matrix $\mathbf{B} = [\mathbf{b}_1^\top; \cdots; \mathbf{b}_n^\top] \in \mathbb{R}^{n \times d}$.

For convenience, we define $\Theta^s(\mathbf{x}^s, \mathbf{y}^s)$ and $\Theta^t(\mathbf{x}^t, \mathbf{y}^t)$ as shown in the following equations according to the Representer theorem,

$$
\begin{aligned}
&\Theta^s(\mathbf{x}^s, \mathbf{y}^s) = \\
&\left[ \left( \textstyle\sum_{l=1}^n \mathbf{b}_l \psi(\mathbf{x}_l)^\top \right) \psi(\mathbf{x}_1^s) \otimes \phi(\mathbf{y}_1^s), \cdots, \left( \textstyle\sum_{l=1}^n \mathbf{b}_l \psi(\mathbf{x}_l)^\top \right) \psi(\mathbf{x}_{n^s}^s) \otimes \phi(\mathbf{y}_{n^s}^s) \right] \in \mathbb{R}^{\infty \times n^s}, \\
&\Theta^t(\mathbf{x}^t, \mathbf{y}^t) = \\
&\left[ \left( \textstyle\sum_{l=1}^n \mathbf{b}_l \psi(\mathbf{x}_l)^\top \right) \psi(\mathbf{x}_1^t) \otimes \phi(\mathbf{y}_1^t), \cdots, \left( \textstyle\sum_{l=1}^n \mathbf{b}_l \psi(\mathbf{x}_l)^\top \right) \psi(\mathbf{x}_{n^t}^t) \otimes \phi(\mathbf{y}_{n^t}^t) \right] \in \mathbb{R}^{\infty \times n^t}.
\end{aligned}
\tag{21}
$$

where $\mathbf{b}_i \in \mathbb{R}^{d \times 1}$ and we definite a new projection matrix $\mathbf{B} = [\mathbf{b}_1^\top; \cdots; \mathbf{b}_n^\top] \in \mathbb{R}^{n \times d}$ ($n = n^s + n^t$). Then, (19) could be rewritten as below,

$$
\begin{aligned}
&\mathbb{D}_{\mathcal{H}} \\
&= \left\Vert \tfrac{1}{n^s} \sum_{i=1}^{n^s} \left( \left( \textstyle\sum_{l=1}^n \mathbf{b}_l \psi(\mathbf{x}_l)^\top \right) \psi(\mathbf{x}_i^s) \otimes \phi(\mathbf{y}_i^s) \right) - \tfrac{1}{n^t} \sum_{j=1}^{n^t} \left( \left( \textstyle\sum_{l=1}^n \mathbf{b}_l \psi(\mathbf{x}_l)^\top \right) \psi(\mathbf{x}_j^t) \otimes \phi(\mathbf{y}_j^t) \right) \right\Vert_{\mathcal{H}}^2 \\
&= \mathrm{tr}\left( \begin{bmatrix} \Theta^s(\mathbf{x}^s, \mathbf{y}^s) & \Theta^t(\mathbf{x}^t, \mathbf{y}^t) \end{bmatrix} \begin{bmatrix} \frac{\mathbf{1}_{n^s \times 1} \mathbf{1}_{n^s \times 1}^\top}{n^s n^s} & \frac{-\mathbf{1}_{n^s \times 1} \mathbf{1}_{n^t \times 1}^\top}{n^s n^t} \\ \frac{-\mathbf{1}_{n^t \times 1} \mathbf{1}_{n^s \times 1}^\top}{n^t n^s} & \frac{\mathbf{1}_{n^t \times 1} \mathbf{1}_{n^t \times 1}^\top}{n^t n^t} \end{bmatrix} \begin{bmatrix} \Theta^s(\mathbf{x}^s, \mathbf{y}^s)^\top \\ \Theta^t(\mathbf{x}^t, \mathbf{y}^t)^\top \end{bmatrix} \right) \\
&= \mathrm{tr}\left( \begin{bmatrix} \Theta^s(\mathbf{x}^s, \mathbf{y}^s)^\top \\ \Theta^t(\mathbf{x}^t, \mathbf{y}^t)^\top \end{bmatrix} \begin{bmatrix} \Theta^s(\mathbf{x}^s, \mathbf{y}^s) & \Theta^t(\mathbf{x}^t, \mathbf{y}^t) \end{bmatrix} \begin{bmatrix} \frac{\mathbf{1}_{n^s \times 1} \mathbf{1}_{n^s \times 1}^\top}{n^s n^s} & \frac{-\mathbf{1}_{n^s \times 1} \mathbf{1}_{n^t \times 1}^\top}{n^s n^t} \\ \frac{-\mathbf{1}_{n^t \times 1} \mathbf{1}_{n^s \times 1}^\top}{n^t n^s} & \frac{\mathbf{1}_{n^t \times 1} \mathbf{1}_{n^t \times 1}^\top}{n^t n^t} \end{bmatrix} \right) \\
&= \mathrm{tr}\left( \begin{bmatrix} \Theta^s(\mathbf{x}^s, \mathbf{y}^s)^\top \Theta^s(\mathbf{x}^s, \mathbf{y}^s) & \Theta^s(\mathbf{x}^s, \mathbf{y}^s)^\top \Theta^t(\mathbf{x}^t, \mathbf{y}^t) \\ \Theta^t(\mathbf{x}^t, \mathbf{y}^t)^\top \Theta^s(\mathbf{x}^s, \mathbf{y}^s) & \Theta^t(\mathbf{x}^t, \mathbf{y}^t)^\top \Theta^t(\mathbf{x}^t, \mathbf{y}^t) \end{bmatrix} \begin{bmatrix} \frac{\mathbf{1}_{n^s \times 1} \mathbf{1}_{n^s \times 1}^\top}{n^s n^s} & \frac{-\mathbf{1}_{n^s \times 1} \mathbf{1}_{n^t \times 1}^\top}{n^s n^t} \\ \frac{-\mathbf{1}_{n^t \times 1} \mathbf{1}_{n^s \times 1}^\top}{n^t n^s} & \frac{\mathbf{1}_{n^t \times 1} \mathbf{1}_{n^t \times 1}^\top}{n^t n^t} \end{bmatrix} \right).
\end{aligned}
\tag{22}
$$

Similar to the proof of Theorem 2, we rewrite $\Theta^s(\mathbf{x}^s, \mathbf{y}^s)^\top \Theta^s(\mathbf{x}^s, \mathbf{y}^s)$, $\Theta^s(\mathbf{x}^s, \mathbf{y}^s)^\top \Theta^t(\mathbf{x}^t, \mathbf{y}^t)$, $\Theta^t(\mathbf{x}^t, \mathbf{y}^t)^\top \Theta^s(\mathbf{x}^s, \mathbf{y}^s)$, $\cdots$ using feature and label kernels. First,

$$
\begin{aligned}
&\Theta^s(\mathbf{x}^s, \mathbf{y}^s)^\top \Theta^s(\mathbf{x}^s, \mathbf{y}^s) \\
&= \begin{bmatrix} \left\langle \left( \textstyle\sum_{l=1}^n \mathbf{b}_l \psi(\mathbf{x}_l)^\top \right) \psi(\mathbf{x}_1^s) \otimes \phi(\mathbf{y}_1^s), \left( \textstyle\sum_{l=1}^n \mathbf{b}_l \psi(\mathbf{x}_l)^\top \right) \psi(\mathbf{x}_1^s) \otimes \phi(\mathbf{y}_1^s) \right\rangle & \cdots & \cdots \\ \left\langle \left( \textstyle\sum_{l=1}^n \mathbf{b}_l \psi(\mathbf{x}_l)^\top \right) \psi(\mathbf{x}_2^s) \otimes \phi(\mathbf{y}_2^s), \left( \textstyle\sum_{l=1}^n \mathbf{b}_l \psi(\mathbf{x}_l)^\top \right) \psi(\mathbf{x}_1^s) \otimes \phi(\mathbf{y}_1^s) \right\rangle & \cdots & \cdots \\ \cdots & \cdots & \cdots \\ \left\langle \left( \textstyle\sum_{l=1}^n \mathbf{b}_l \psi(\mathbf{x}_l)^\top \right) \psi(\mathbf{x}_{n^s}^s) \otimes \phi(\mathbf{y}_{n^s}^s), \left( \textstyle\sum_{l=1}^n \mathbf{b}_l \psi(\mathbf{x}_l)^\top \right) \psi(\mathbf{x}_1^s) \otimes \phi(\mathbf{y}_1^s) \right\rangle & \cdots & \cdots \end{bmatrix},
\end{aligned}
\tag{23}
$$

where $\langle \bullet, \bullet \rangle$ denotes the inner product between two vectors. Moreover, we have the following equation,

$$\left\langle \left( \sum_{l=1}^{n} \mathbf{b}_l \psi(\mathbf{x}_l)^\top \right) \psi(\mathbf{x}_i^s) \otimes \phi(\mathbf{y}_j^s), \left( \sum_{l=1}^{n} \mathbf{b}_l \psi(\mathbf{x}_l)^\top \right) \psi(\mathbf{x}_k) \otimes \phi(\mathbf{y}_m) \right\rangle$$

$$= \left( \left( \sum_{l=1}^{n} \mathbf{b}_l \psi(\mathbf{x}_l)^\top \right) \psi(\mathbf{x}_i^s) \otimes \phi(\mathbf{y}_j^s) \right)^\top \left( \left( \sum_{l=1}^{n} \mathbf{b}_l \psi(\mathbf{x}_l)^\top \right) \psi(\mathbf{x}_k) \otimes \phi(\mathbf{y}_m) \right)$$

$$= \left( \psi(\mathbf{x}_i^s)^\top \left( \sum_{l=1}^{n} \psi(\mathbf{x}_l) \mathbf{b}_l^\top \right) \otimes \phi(\mathbf{y}_j^s)^\top \right) \left( \left( \sum_{l=1}^{n} \mathbf{b}_l \psi(\mathbf{x}_l)^\top \right) \psi(\mathbf{x}_k) \otimes \phi(\mathbf{y}_m) \right)$$

$$= \left( \psi(\mathbf{x}_i^s)^\top \left( \sum_{l=1}^{n} \psi(\mathbf{x}_l) \mathbf{b}_l^\top \right) \left( \sum_{l=1}^{n} \mathbf{b}_l \psi(\mathbf{x}_l)^\top \right) \psi(\mathbf{x}_k) \right) \otimes \left( \phi(\mathbf{y}_j^s)^\top \phi(\mathbf{y}_m) \right)$$

$$= \left[ k^X(\mathbf{x}_i^s, \mathbf{x}_1), k^X(\mathbf{x}_i^s, \mathbf{x}_2), \cdots, k^X(\mathbf{x}_i^s, \mathbf{x}_n) \right] \mathbf{B} \mathbf{B}^\top \left[ k^X(\mathbf{x}_1, \mathbf{x}_k), k^X(\mathbf{x}_2, \mathbf{x}_k), \cdots, k^X(\mathbf{x}_n, \mathbf{x}_k) \right]^\top \otimes k^Y(\mathbf{y}_j^s, \mathbf{y}_m)$$

$$= \left[ k^X(\mathbf{x}_i^s, \mathbf{x}_1), k^X(\mathbf{x}_i^s, \mathbf{x}_2), \cdots, k^X(\mathbf{x}_i^s, \mathbf{x}_n) \right] \mathbf{B} \mathbf{B}^\top \left[ k^X(\mathbf{x}_1, \mathbf{x}_k), k^X(\mathbf{x}_2, \mathbf{x}_k), \cdots, k^X(\mathbf{x}_n, \mathbf{x}_k) \right]^\top k^Y(\mathbf{y}_j^s, \mathbf{y}_m)$$

$$= \mathbf{K}_{(i,\bullet)}^{XX} \mathbf{B} \mathbf{B}^\top \mathbf{K}_{(\bullet,k)}^{XX} k^Y(\mathbf{y}_j, \mathbf{y}_m),$$

(24)

where the subscripts $(i, \bullet)$ and $(\bullet, k)$ denote the i-th row vector and the k-th column vector of a given matrix, respectively. Then, we can obtain the following equation,

$$\Theta^s(\mathbf{x}^s, \mathbf{y}^s)^\top \Theta^s(\mathbf{x}^s, \mathbf{y}^s) =$$

$$\begin{bmatrix} \mathbf{K}_{(1,\bullet)}^{XX} \mathbf{B} \mathbf{B}^\top \mathbf{K}_{(\bullet,1)}^{XX} k^Y(\mathbf{y}_1, \mathbf{y}_1) & \cdots & \mathbf{K}_{(1,\bullet)}^{XX} \mathbf{B} \mathbf{B}^\top \mathbf{K}_{(\bullet,n^s)}^{XX} k^Y(\mathbf{y}_1, \mathbf{y}_{n^s}) \\ \mathbf{K}_{(2,\bullet)}^{XX} \mathbf{B} \mathbf{B}^\top \mathbf{K}_{(\bullet,1)}^{XX} k^Y(\mathbf{y}_2, \mathbf{y}_1) & \cdots & \mathbf{K}_{(2,\bullet)}^{XX} \mathbf{B} \mathbf{B}^\top \mathbf{K}_{(\bullet,n^s)}^{XX} k^Y(\mathbf{y}_2, \mathbf{y}_{n^s}) \\ \cdots & \cdots & \cdots \\ \mathbf{K}_{(n^s,\bullet)}^{XX} \mathbf{B} \mathbf{B}^\top \mathbf{K}_{(\bullet,1)}^{XX} k^Y(\mathbf{y}_{n^s}, \mathbf{y}_1) & \cdots & \mathbf{K}_{(n^s,\bullet)}^{XX} \mathbf{B} \mathbf{B}^\top \mathbf{K}_{(\bullet,n^s)}^{XX} k^Y(\mathbf{y}_{n^s}, \mathbf{y}_{n^s}) \end{bmatrix}.$$

(25)

Similarly, $\Theta^t(\mathbf{x}^t, \mathbf{y}^t)^\top \Theta^t(\mathbf{x}^t, \mathbf{y}^t) =$

$$\begin{bmatrix} \mathbf{K}_{(n^s+1,\bullet)}^{XX} \mathbf{B} \mathbf{B}^\top \mathbf{K}_{(\bullet,n^s+1)}^{XX} k^Y(\mathbf{y}_{n^s+1}, \mathbf{y}_{n^s+1}) & \cdots & \mathbf{K}_{(n^s+1,\bullet)}^{XX} \mathbf{B} \mathbf{B}^\top \mathbf{K}_{(\bullet,n)}^{XX} k^Y(\mathbf{y}_{n^s+1}, \mathbf{y}_n) \\ \mathbf{K}_{(n^s+2,\bullet)}^{XX} \mathbf{B} \mathbf{B}^\top \mathbf{K}_{(\bullet,n^s+1)}^{XX} k^Y(\mathbf{y}_{n^s+2}, \mathbf{y}_{n^s+1}) & \cdots & \mathbf{K}_{(n^s+2,\bullet)}^{XX} \mathbf{B} \mathbf{B}^\top \mathbf{K}_{(\bullet,n)}^{XX} k^Y(\mathbf{y}_{n^s+2}, \mathbf{y}_n) \\ \cdots & \cdots & \cdots \\ \mathbf{K}_{(n,\bullet)}^{XX} \mathbf{B} \mathbf{B}^\top \mathbf{K}_{(\bullet,n^s+1)}^{XX} k^Y(\mathbf{y}_n, \mathbf{y}_{n^s+1}) & \cdots & \mathbf{K}_{(n,\bullet)}^{XX} \mathbf{B} \mathbf{B}^\top \mathbf{K}_{(\bullet,n)}^{XX} k^Y(\mathbf{y}_n, \mathbf{y}_n) \end{bmatrix}.$$

(26)

$\Theta^s(\mathbf{x}^s, \mathbf{y}^s)^\top \Theta^t(\mathbf{x}^t, \mathbf{y}^t) =$

$$\begin{bmatrix} \mathbf{K}_{(1,\bullet)}^{XX} \mathbf{B} \mathbf{B}^\top \mathbf{K}_{(\bullet,n^s+1)}^{XX} k^Y(\mathbf{y}_1, \mathbf{y}_{n^s+1}) & \cdots & \mathbf{K}_{(1,\bullet)}^{XX} \mathbf{B} \mathbf{B}^\top \mathbf{K}_{(\bullet,n)}^{XX} k^Y(\mathbf{y}_1, \mathbf{y}_n) \\ \mathbf{K}_{(2,\bullet)}^{XX} \mathbf{B} \mathbf{B}^\top \mathbf{K}_{(\bullet,n^s+1)}^{XX} k^Y(\mathbf{y}_2, \mathbf{y}_{n^s+1}) & \cdots & \mathbf{K}_{(2,\bullet)}^{XX} \mathbf{B} \mathbf{B}^\top \mathbf{K}_{(\bullet,n)}^{XX} k^Y(\mathbf{y}_2, \mathbf{y}_n) \\ \cdots & \cdots & \cdots \\ \mathbf{K}_{(n^s,\bullet)}^{XX} \mathbf{B} \mathbf{B}^\top \mathbf{K}_{(\bullet,n^s+1)}^{XX} k^Y(\mathbf{y}_{n^s}, \mathbf{y}_{n^s+1}) & \cdots & \mathbf{K}_{(n^s,\bullet)}^{XX} \mathbf{B} \mathbf{B}^\top \mathbf{K}_{(\bullet,n)}^{XX} k^Y(\mathbf{y}_{n^s}, \mathbf{y}_n) \end{bmatrix}.$$

(27)

$\Theta^t(\mathbf{x}^t, \mathbf{y}^t)^\top \Theta^s(\mathbf{x}^s, \mathbf{y}^s) =$

$$
\begin{bmatrix}
\mathbf{K}^{XX}_{(n^s+1,\bullet)}\mathbf{B}\mathbf{B}^\top \mathbf{K}^{XX}_{(\bullet,1)}k^Y(\mathbf{y}_{n^s+1},\mathbf{y}_1) & \cdots & \mathbf{K}^{XX}_{(n^s+1,\bullet)}\mathbf{B}\mathbf{B}^\top \mathbf{K}^{XX}_{(\bullet,n^s)}k^Y(\mathbf{y}_{n^s+1},\mathbf{y}_{n^s}) \\[2mm]
\mathbf{K}^{XX}_{(n^s+2,\bullet)}\mathbf{B}\mathbf{B}^\top \mathbf{K}^{XX}_{(\bullet,1)}k^Y(\mathbf{y}_{n^s+2},\mathbf{y}_1) & \cdots & \mathbf{K}^{XX}_{(n^s+2,\bullet)}\mathbf{B}\mathbf{B}^\top \mathbf{K}^{XX}_{(\bullet,n^s)}k^Y(\mathbf{y}_{n^s+2},\mathbf{y}_{n^s}) \\[2mm]
\cdots & \cdots & \cdots \\[2mm]
\mathbf{K}^{XX}_{(n,\bullet)}\mathbf{B}\mathbf{B}^\top \mathbf{K}^{XX}_{(\bullet,1)}k^Y(\mathbf{y}_n,\mathbf{y}_1) & \cdots & \mathbf{K}^{XX}_{(n,\bullet)}\mathbf{B}\mathbf{B}^\top \mathbf{K}^{XX}_{(\bullet,n^s)}k^Y(\mathbf{y}_n,\mathbf{y}_{n^s})
\end{bmatrix}. \tag{28}
$$

According to (25) $\sim$ (28), we could obtain the following equation,

$$
\mathbb{D}_{\mathcal{H}} = \mathrm{tr}\Bigg(
\begin{bmatrix}
\Theta^s(\mathbf{x}^s,\mathbf{y}^s)^\top \Theta^s(\mathbf{x}^s,\mathbf{y}^s) & \Theta^s(\mathbf{x}^s,\mathbf{y}^s)^\top \Theta^t(\mathbf{x}^t,\mathbf{y}^t) \\[2mm]
\Theta^t(\mathbf{x}^t,\mathbf{y}^t)^\top \Theta^s(\mathbf{x}^s,\mathbf{y}^s) & \Theta^t(\mathbf{x}^t,\mathbf{y}^t)^\top \Theta^t(\mathbf{x}^t,\mathbf{y}^t)
\end{bmatrix}
\begin{bmatrix}
\frac{\mathbf{1}_{n^s\times 1}\mathbf{1}^\top_{n^s\times 1}}{n^s n^s} & \frac{-\mathbf{1}_{n^s\times 1}\mathbf{1}^\top_{n^t\times 1}}{n^s n^t} \\[2mm]
\frac{-\mathbf{1}_{n^t\times 1}\mathbf{1}^\top_{n^s\times 1}}{n^t n^s} & \frac{\mathbf{1}_{n^t\times 1}\mathbf{1}^\top_{n^t\times 1}}{n^t n^t}
\end{bmatrix}
\Bigg)
$$

$$
= \mathrm{tr}\Big(\mathbf{B}^\top \mathbf{K}^{XX}(\mathbf{K}^{YY}\odot \mathbf{M}^J)\mathbf{K}^{XX}\mathbf{B}\Big). \tag{29}
$$

$\square$

## C  Probability Distribution Distances

### C.1  Maximum Mean Discrepancy

The maximum mean discrepancy (MMD) [12] establishes the mean embedding of the marginal probability distribution in a RKHS endowed by the kernel $k^X$ (feature mapping $\psi$), and using finite samples to empirically estimate the distance between $\mu_{X^s}$ (mean embedding of source domain) and $\mu_{X^t}$ (mean embedding of target domain) with the Hilbert-Schmidt norm as the following equation,

$$
\mathbb{D}_{\mathcal{H}}\Big(\mathcal{P}^s(\mathbf{X}^s),\mathcal{P}^t(\mathbf{X}^t)\Big) = \left\|\mathbb{E}\Big(\psi(\mathbf{X}^s)\Big) - \mathbb{E}\Big(\psi(\mathbf{X}^t)\Big)\right\|^2_{\mathcal{H}} = \left\|\mu_{X^s} - \mu_{X^t}\right\|^2_{\mathcal{H}}
$$

$$
= \left\|\tfrac{1}{n^s}\sum_{i=1}^{n^s}\psi(\mathbf{x}_i) - \tfrac{1}{n^t}\sum_{j=1}^{n^t}\psi(\mathbf{x}_j)\right\|^2_{\mathcal{H}} = \mathrm{tr}(\mathbf{K}^{XX}\mathbf{M}^M), \tag{30}
$$

where $\mathbf{K}^{XX} = \psi(\mathbf{X})^\top \psi(\mathbf{X}) \in \mathbb{R}^{n\times n}$ and $k^{XX}_{ij} = k^X(\mathbf{x}_i,\mathbf{x}_j)$. Besides, the MMD matrix $\mathbf{M}^M$ can be computed as below,

$$
m^M_{ij} = \begin{cases}
1/(n^s n^s), & \mathbf{x}_i,\mathbf{x}_j \in \mathcal{D}^s \\[2mm]
1/(n^t n^t), & \mathbf{x}_i,\mathbf{x}_j \in \mathcal{D}^t \\[2mm]
-1/(n^s n^t), & \text{otherwise.}
\end{cases} \tag{31}
$$

Moreover, the MMD in a projected RKHS is $\mathrm{tr}(\mathbf{B}^\top \mathbf{K}^{XX}\mathbf{M}^m\mathbf{K}^{XX}\mathbf{B})$.

### C.2  Class-Wise Maximum Mean Discrepancy

The class-wise maximum mean discrepancy (CMMD) [20] constructs the sum of MMD for each specific class as the following equation,

$$\mathbb{D}_{\mathcal{H}}\Big(\mathcal{P}^s(\mathbf{X}^s|\mathbf{Y}^s), \mathcal{P}^t(\mathbf{X}^t|\mathbf{Y}^t)\Big)$$

$$= \sum_{c=1}^{C} \left\lVert \mathbb{E}\Big(\psi(\mathbf{X}^{s,c})\Big) - \mathbb{E}\Big(\psi(\mathbf{X}^{t,c})\Big) \right\rVert_{\mathcal{H}}^2 = \sum_{c=1}^{C} \left\lVert \mu_{X^{s,c}} - \mu_{X^{t,c}} \right\rVert_{\mathcal{H}}^2 \tag{32}$$

$$= \sum_{c=1}^{C} \left\lVert \frac{1}{n^{s,c}} \sum_{i=1}^{n^{s,c}} \psi(\mathbf{x}_i) - \frac{1}{n^{t,c}} \sum_{j=1}^{n^{t,c}} \psi(\mathbf{x}_j) \right\rVert_{\mathcal{H}}^2 = \sum_{c=1}^{C} \mathrm{tr}(\mathbf{K}^{XX}\mathbf{M}^{C,c}),$$

where the MMD matrix $\mathbf{M}^{C,c}$ can be computed as below,

$$m_{ij}^{C,c} = \begin{cases} 1/(n^{s,c}n^{s,c}), & \mathbf{x}_i \in \mathcal{D}^{s,c}, \mathbf{x}_j \in \mathcal{D}^{s,c} \\[2mm] 1/(n^{t,c}n^{t,c}), & \mathbf{x}_i \in \mathcal{D}^{t,c}, \mathbf{x}_j \in \mathcal{D}^{t,c} \\[2mm] -1/(n^{s,c}n^{t,c}), & \mathbf{x}_i \in \mathcal{D}^{s,c}, \mathbf{x}_j \in \mathcal{D}^{t,c} \\[2mm] -1/(n^{t,c}n^{s,c}), & \mathbf{x}_i \in \mathcal{D}^{t,c}, \mathbf{x}_j \in \mathcal{D}^{s,c} \\[2mm] 0, & \text{otherwise.} \end{cases} \tag{33}$$

Similarly, the CMMD in a projected RKHS is $\sum_{c=1}^{C} \mathrm{tr}(\mathbf{B}^\top \mathbf{K}^{XX}\mathbf{M}^{C,c}\mathbf{K}^{XX}\mathbf{B})$.

### C.3 Weighted Class-Wise Maximum Mean Discrepancy

To deal with class imbalanced dataset, the weighted class-wise maximum mean discrepancy (WCMMD) introduces the class prior probability $\mathcal{P}(\mathbf{Y})$ into the CMMD [21], which pays more attention on the large-size categories and is formulated as the following equation,

$$\sum_{c=1}^{C} \left\lVert \frac{\mathcal{P}^s(\mathbf{y}^s=c)}{n^{s,c}} \sum_{i=1}^{n^{s,c}} \psi(\mathbf{x}_i) - \frac{\mathcal{P}^t(\mathbf{y}^t=c)}{n^{t,c}} \sum_{j=1}^{n^{t,c}} \psi(\mathbf{x}_j) \right\rVert_{\mathcal{H}}^2 \tag{34}$$

$$= \sum_{c=1}^{C} \left\lVert \frac{1}{n^s} \sum_{i=1}^{n^{s,c}} \psi(\mathbf{x}_i) - \frac{1}{n^t} \sum_{j=1}^{n^{t,c}} \psi(\mathbf{x}_j) \right\rVert_{\mathcal{H}}^2 = \sum_{c=1}^{C} \mathrm{tr}(\mathbf{K}^{XX}\mathbf{M}^{WC,c}),$$

where $\mathbf{M}^{WC,c}$ can be computed with the following equation,

$$m_{ij}^{WC,c} = \begin{cases} 1/(n^s n^s), & \mathbf{x}_i \in \mathcal{D}^{s,c}, \mathbf{x}_j \in \mathcal{D}^{s,c} \\[2mm] 1/(n^t n^t), & \mathbf{x}_i \in \mathcal{D}^{t,c}, \mathbf{x}_j \in \mathcal{D}^{t,c} \\[2mm] -1/(n^s n^t), & \mathbf{x}_i \in \mathcal{D}^{s,c}, \mathbf{x}_j \in \mathcal{D}^{t,c} \\[2mm] -1/(n^t n^s), & \mathbf{x}_i \in \mathcal{D}^{t,c}, \mathbf{x}_j \in \mathcal{D}^{s,c} \\[2mm] 0, & \text{otherwise.} \end{cases} \tag{35}$$

Similarly, the WCMMD in a projected RKHS is $\sum_{c=1}^{C} \mathrm{tr}(\mathbf{B}^\top \mathbf{K}^{XX}\mathbf{M}^{WC,c}\mathbf{K}^{XX}\mathbf{B})$.

## D  The Uniformity of JMMD

**Theorem 5** *The marginal, class conditional and weighted class conditional probability distribution distances are three special cases of JMMD with label reproducing kernels $\mathbf{K}^1$, $\mathbf{K}^2$ and $\mathbf{K}^3$. $\mathbf{K}^1 = \mathbf{1}_{n \times n}$ is a matrix whose elements are all 1 with the size of $n \times n$, and $\mathbf{K}^2$, $\mathbf{K}^3$ are defined as below,*

$$
k_{ij}^2 = \begin{cases}
(n^s n^s)/(n^{s,c} n^{s,c}), & \mathbf{x}_i \in \mathcal{D}^{s,c}, \mathbf{x}_j \in \mathcal{D}^{s,c} \\
(n^t n^t)/(n^{t,c} n^{t,c}), & \mathbf{x}_i \in \mathcal{D}^{t,c}, \mathbf{x}_j \in \mathcal{D}^{t,c} \\
(n^s n^t)/(n^{s,c} n^{t,c}), & \mathbf{x}_i \in \mathcal{D}^{s,c}, \mathbf{x}_j \in \mathcal{D}^{t,c} \\
(n^t n^s)/(n^{t,c} n^{s,c}), & \mathbf{x}_i \in \mathcal{D}^{t,c}, \mathbf{x}_j \in \mathcal{D}^{s,c} \\
0, & \text{otherwise},
\end{cases}
\tag{36}
$$

$$
k_{ij}^3 = \begin{cases}
1, & \mathbf{x}_i \in D^{s,c}, \mathbf{x}_j \in \mathcal{D}^{s,c} \\
1, & \mathbf{x}_i \in D^{t,c}, \mathbf{x}_j \in \mathcal{D}^{t,c} \\
1, & \mathbf{x}_i \in D^{s,c}, \mathbf{x}_j \in \mathcal{D}^{t,c} \\
1, & \mathbf{x}_i \in D^{t,c}, \mathbf{x}_j \in \mathcal{D}^{s,c} \\
0, & \text{otherwise},
\end{cases}
\tag{37}
$$

where the superscript 's/t,c' denotes data points from the c-th class in the source/target domain.

*Proof:*

As proved before, the formulations of concise JMMD are $\mathrm{tr}(\mathbf{K}^{XX}(\mathbf{K}^{YY} \odot \mathbf{M}^J))$ (in a RKHS) and $\mathrm{tr}(\mathbf{B}^\top \mathbf{K}^{XX}(\mathbf{K}^{YY} \odot \mathbf{M}^J)\mathbf{K}^{XX}\mathbf{B})$ (in a projected RKHS). Moreover, the formulations of marginal probability distribution distance are $\mathrm{tr}(\mathbf{K}^{XX}\mathbf{M}^M)$ (in a RKHS) and $\mathrm{tr}(\mathbf{B}^\top \mathbf{K}^{XX}\mathbf{M}^M\mathbf{K}^{XX}\mathbf{B})$ (in a projected RKHS). The formulations of class conditional probability distribution distance are $\mathrm{tr}(\mathbf{K}^{XX}\mathbf{M}^{C,c})$ (in a RKHS) and $\mathrm{tr}(\mathbf{B}^\top \mathbf{K}^{XX}\mathbf{M}^{C,c}\mathbf{K}^{XX}\mathbf{B})$ (in a projected RKHS). The formulations of weighted class conditional probability distribution distance are $\mathrm{tr}(\mathbf{K}^{XX}\mathbf{M}^{WC,c})$ (in a RKHS) and $\mathrm{tr}(\mathbf{B}^\top \mathbf{K}^{XX}\mathbf{M}^{WC,c}\mathbf{K}^{XX}\mathbf{B})$ (in a projected RKHS). It is easy to verify that $\mathbf{K}^1 \odot \mathbf{M}^J = \mathbf{M}^M$, $\mathbf{K}^2 \odot \mathbf{M}^J = \mathbf{M}^{C,c}$ and $\mathbf{K}^3 \odot \mathbf{M}^J = \mathbf{M}^{WC,c}$. Therefore, the marginal, class conditional and weighted class conditional probability distribution distances are three special cases of JMMD with different label reproducing kernels $\mathbf{K}^1$, $\mathbf{K}^2$ and $\mathbf{K}^3$. We will prove $\mathbf{K}^1$, $\mathbf{K}^2$ and $\mathbf{K}^3$ are the reproducing kernels in next Subsection.

$\square$

# E   Reproducing Kernels

**Theorem 6** $K^1$, $K^2$, $K^3$ and $K^4$ *are the reproducing kernels, where* $K^1$, $K^2$, $K^3$ *are defined in Theorem 5.*

*Proof:*

$\mathbf{K}^1$, $\mathbf{K}^2$ and $\mathbf{K}^3$: According to the Mercer's theorem [23], we only have to prove that the Gram matrices $\mathbf{G}^1$, $\mathbf{G}^2$ and $\mathbf{G}^3$ corresponding to $\mathbf{K}^1$, $\mathbf{K}^2$ and $\mathbf{K}^3$ are semi-positive definite matrices. In fact, the Gram matrices $\mathbf{G}^2$ and $\mathbf{G}^3$ can be decomposed into the following equation,

$$
\mathbf{G}^2 = \sum_{c=1}^C \mathbf{G}^{2,c}, \quad \mathbf{G}^3 = \sum_{c=1}^C \mathbf{G}^{3,c}.
\tag{38}
$$

It is obvious that the sum of several semi-positive definite matrices is also a semi-positive definite matrix, thus we only have to prove that the Gram matrices $\mathbf{G}^1$, $\mathbf{G}^{2,c}$ and $\mathbf{G}^{3,c}$ are semi-positive definite. $\mathbf{G}^1$, $\mathbf{G}^{2,c}$ and $\mathbf{G}^{3,c}$ can be decomposed into the following equations,

$$
\mathbf{G}^1 = \mathbf{p}^1 \mathbf{p}^{1\top}, \ \mathbf{G}^{2,c} = \mathbf{p}^{2,c} \mathbf{p}^{2,c\top}, \ \mathbf{G}^{3,c} = \mathbf{p}^{3,c} \mathbf{p}^{3,c\top},
\tag{39}
$$

where $\mathbf{p}^1 = \mathbf{1}_n$ is a column vector whose elements are all 1, and $\mathbf{p}^{2,c} \in \mathbb{R}^n$, $\mathbf{p}^{3,c} \in \mathbb{R}^n$ could be defined as below,

$$p_i^{2,c} = \begin{cases} n^s/n^{s,c}, & \mathbf{x}_i \in \mathcal{D}^{s,c} \\[2mm] n^t/n^{t,c}, & \mathbf{x}_i \in \mathcal{D}^{t,c} \\[2mm] 0, & \text{otherwise,} \end{cases} \tag{40}$$

$$p_i^{3,c} = \begin{cases} 1, & \mathbf{x}_i \in \mathcal{D}^{s,c} \\[2mm] 1, & \mathbf{x}_i \in \mathcal{D}^{t,c} \\[2mm] 0, & \text{otherwise,} \end{cases} \tag{41}$$

where $p_i^{2/3,c}$ is the value of the i-th component of $\mathbf{l}^{2/3,c}$. For $\forall \mathbf{x} \in \mathbb{R}^n$ and $\mathbf{x} \neq 0$, we have,

$$\mathbf{x}^\top \mathbf{G}^1 \mathbf{x} = \mathbf{x}^\top \mathbf{p}^1 \mathbf{p}^{1\top} \mathbf{x} = \mathbf{x}^\top \mathbf{1}_n \mathbf{1}_n^\top \mathbf{x}$$
$$= (x_1 + x_2 + ... + x_n)^2 \geq 0, \tag{42}$$

$$\mathbf{x}^\top \mathbf{G}^{2,c} \mathbf{x} = \mathbf{x}^\top \mathbf{p}^{2,c} \mathbf{p}^{2,c\top} \mathbf{x}$$
$$= (x_1 p_1^{2,c} + x_2 p_2^{2,c} + ... + x_n p_n^{2,c})^2 \geq 0. \tag{43}$$

and,

$$\mathbf{x}^\top \mathbf{G}^{3,c} \mathbf{x} = \mathbf{x}^\top \mathbf{p}^{3,c} \mathbf{p}^{3,c\top} \mathbf{x}$$
$$= (x_1 p_1^{3,c} + x_2 p_2^{3,c} + \cdots + x_n p_n^{3,c})^2 \geq 0. \tag{44}$$

Therefore, $\mathbf{G}^1$, $\mathbf{G}^2$ and $\mathbf{G}^3$ are semi-positive definite matrices. Then, $\mathbf{K}^1$, $\mathbf{K}^2$ and $\mathbf{K}^3$ are the reproducing kernels.

$\mathbf{K}^4$: For $\forall \mathbf{x} \in \mathbb{R}^n$ and $\mathbf{x} \neq 0$, due to $w_{ij} \geq 0$, we have,

$$\mathbf{x}^\top \mathbf{G}^4 \mathbf{x} = \sum_{i=1}^n \sum_{j=1}^n w_{ij}(x_i - x_j)^2 \geq 0, \tag{45}$$

where $\mathbf{G}^4$ is the Gram matrix of $\mathbf{K}^4$ and $\mathbf{W}$ is defined as below,

$$w_{ij} = \begin{cases} (n^s n^s)/(n^{s,c} n^{s,c}), & \mathbf{x}_i \in \mathcal{D}^{s,c}, \mathbf{x}_j \in \mathcal{D}^{s,c} \\[2mm] (n^t n^t)/(n^{t,c} n^{t,c}), & \mathbf{x}_i \in \mathcal{D}^{t,c}, \mathbf{x}_j \in \mathcal{D}^{t,c} \\[2mm] 0, & \text{otherwise.} \end{cases} \tag{46}$$

Therefore, $\mathbf{G}^4$ is a semi-positive matrix and $\mathbf{K}^4$ is the reproducing kernel.

$\square$

# F  Experiments

We run the JDA+JMMD/HSIC/Our(JMMD-HSIC) and adopt the classifiers of 1-nearest neighbor (1-NN), support vector machines (SVM[*]), label propagation (LP[†]) [60] and nearest class prototype

---

[*]https://www.csie.ntu.edu.tw/ cjlin/libsvm
[†]https://www.escience.cn/people/fpnie/index.html

Table 5: Ablation study using different classifiers/labels on the Office10-Caltech10 dataset with SURF features.

| Classifier | 1-NN | SVM | | LP | | NCP | |
|---|---|---|---|---|---|---|---|
| Original Features | 40.9 | 47.7 | | 48.3 | | 45.7 | |
| Label | Hard | Hard | Soft | Hard | Soft | Hard | Soft |
| JDA+JMMD | 47.7 | 50.2 | 49.2 | 54.2 | 52.8 | 47.8 | 41.4 |
| JDA+HSIC | 47.9 | 49.0 | 48.5 | 54.0 | 52.8 | 48.8 | 46.8 |
| JDA+Our | **49.6** | **50.9** | **49.9** | **55.4** | **54.3** | **49.6** | **47.1** |

Table 6: Comparison average results of our proposed SPL+JMMD-HSIC with state-of-the-art DA methods on Office10-Caltech10 dataset with DECAF-6 features. A, C, D, W in the second row denotes domains of Amazon, Caltech, Dslr, and Webcam, respectively.

| Source | Venue | Amazon | | | Caltech | | | Dslr | | | Webcam | | | Avg. |
|---|---|---|---|---|---|---|---|---|---|---|---|---|---|---|
| Target | | C | D | W | A | D | W | A | C | W | A | C | D | |
| PGCD [54] | TIP'23 | 86.5 | 90.4 | 84.1 | 92.5 | 92.4 | 91.2 | 92.5 | 87.6 | 100.0 | 91.6 | 85.3 | 100.0 | 91.2 |
| RMMD-II [55] | TNNLS'23 | 88.4 | 91.7 | 92.9 | 93.4 | 96.8 | 95.9 | 93.6 | 88.9 | 100.0 | 92.2 | 88.9 | 100.0 | 93.6 |
| SPL [50] | AAAI'20 | 87.4 | 89.2 | 95.3 | 92.7 | 98.7 | 93.2 | 92.9 | 88.6 | 98.6 | 92.0 | 87.0 | 100.0 | 93.0 |
| SPL+Our | - | 90.0 | 96.8 | 93.9 | 93.7 | 99.4 | 93.9 | 93.8 | 90.3 | 100.0 | 93.3 | 89.4 | 99.4 | 94.5 |
| OGL$^2$P [51] | TIP'25 | 89.7 | 97.5 | 91.9 | 94.3 | 98.7 | 95.9 | 94.2 | 90.2 | 99.3 | 94.6 | 89.5 | 100.0 | 94.6 |
| OGL$^2$P+Our | - | 90.2 | 97.8 | 93.2 | 95.3 | 99.2 | 95.7 | 94.6 | 90.5 | 100.0 | 94.2 | 89.3 | 100.0 | **95.0** |

Table 7: Comparison of average results of our proposed SPL+JMMD-HSIC with state-of-the-art DA methods on ImageCLEF-DA dataset with ResNet-50 features. C, I, P in the second row denotes domains of Caltech-256, ImageNet ILSVRC, and Pascal VOC, respectively.

| Source | Venue | Caltech -256 | | ImageNet ILSVRC | | Pascal VOC | | Avg. |
|---|---|---|---|---|---|---|---|---|
| Target | | I | P | C | P | C | I | |
| RMMD-I [55] | TNNLS'23 | 93.2 | 78.3 | 95.7 | 79.5 | 95.5 | 92.0 | 89.0 |
| RSDA-MSTN [58] | TPAMI'24 | 93.3 | 79.3 | 97.8 | 80.5 | 96.8 | 94.2 | 90.3 |
| SPL [50] | AAAI'20 | 95.7 | 80.5 | 96.7 | 78.3 | 96.3 | 94.5 | 90.3 |
| SPL+Our | - | 96.3 | 81.4 | 96.7 | 80.5 | 96.7 | 95.0 | 91.1 |
| OGL$^2$P [51] | TIP'25 | 95.8 | 81.2 | 96.8 | 82.2 | 97.2 | 95.7 | 91.5 |
| OGL$^2$P+Our | - | 96.5 | 81.8 | 97.4 | 83.5 | 97.8 | 96.0 | **92.2** |

(NCP[‡]) [50] on the Office-Caltech10 dataset with SURF features (average classification results on 12 DA tasks). As can be seen from Tab. 5, JMMD and HSIC perform better than the original features as JMMD matches the distributions of the source domain and target domain, and HSIC enhances domain-specific discriminative structures. The proposed JMMD-HSIC could achieve the best results no matter what classifiers or labels are, which shows the effectiveness of JMMD-HSIC and indicates that it is necessary to jointly consider JMMD and HSIC for a better DA capacity. Here, the symbol 'Soft' denotes the probability soft label and the symbol 'Hard' is the hard (one-hot) label. Notably, 1-NN could not produce a soft label thus only 'Hard' is reported. It can be seen that the performance of the 'Soft' label is even worse than that of the 'Hard' label, and it may be because the performance heavily depends on the quality of predicted soft labels of the target domain [62, 63].

We compare our proposed approach with existing state-of-the-art shallow (SPL [50], PGCD [54], RMMD [55]) and deep DA approaches (RSDA-MSTN [58], OGL$^2$P [51]) on $D^1$ and $D^2$. As can be seen from Tabs. 6 and 7, our proposed approach is better than the baseline methods SPL and OGL$^2$P on average, and has achieved 1.5%/0.4% and 0.8/0.7% improvements on the two datasets, respectively. Besides, OGL$^2$P+JMMD-HSIC could achieve the best average results among all compared approaches, which has achieved 0.4% and 0.7% improvements compared with the second-best methods, *i.e.*, OGL$^2$P. Generally speaking, these results can show the effectiveness and competitiveness of our proposed JMMD-HSIC.

---

[‡]https://github.com/hellowangqian/domainadaptation-capls

