# OpenReview forum: "Rethinking Joint Maximum Mean Discrepancy for Visual Domain Adaptation"
_NeurIPS.cc/2025/Conference — NeurIPS 2025 oral_

### Official Review · Reviewer_J6vh · 2025-06-19

**Clarity:** 2
**Significance:** 3
**Originality:** 3
**Rating:** 5
**Confidence:** 3

**Summary:**

This paper conducts research on a JMMD, joint maximum mean discrepancy.
They mention that the empirical estimation of JMMD includes a tensor-product operator, which makes it difficult to obtain the partial derivative.
They conduct a theoretical analysis and have two findings: 1) the uniformity of JMMD, 2) similarity weights, which strengthen the intra-class compactness in the graph of Hilbert Schmidt independence criterion (HSIC), take opposite signs in the graph of JMMD, revealing the reason why JMMD degrades the feature discrimination.
With these findings, they propose a novel loss, JMMD-HSIC.
Extensive results prove the effectiveness of the proposed method.

**Questions:**

See weakness 2-4.

**Ethical Concerns:**

["NO or VERY MINOR ethics concerns only"]

**Final Justification:**

After reading the rebuttal and other reviews, I think this paper can be accepted. I raise my score to 5.

**Limitations:**

No. I think the authors can have discussions on the limitations of the proposed loss, maybe it will unveil other important factors in domain adaptation.

**Quality:**

4

**Strengths And Weaknesses:**

Pros:
1. This paper pays attention to an interesting problem, JMMD. JMMD is a very famous distance in the domain adaptation research community. The authors reflect on the limitations of this classical distance, which is new to me.
2. Avoiding tensor-product operators helps obtain the partial derivative. I think this kind of modification is good.
3. Now we know that some domain alignment methods will hurt discriminability, but we have not found the root cause. I think this paper opens a good topic.
4. I think it is good and interesting to tackle domain adaptation from a theoretical perspective.

Cons:
1. I think the paper can be polished, especially the abs and introduction section. The authors can use some bold texts to highlight key ideas.
2. I think the novelty of this paper is ok for me, but not very high. The authors can have more discussions on the novelty of the paper.
3. I want to see more extensive expeirment results, eg. combination with more methods.
4. The authors can have discussions on the limitations or failure cases of the proposed method.

---

> ### Author Rebuttal · Authors · 2025-07-30
>
> **Weaknesses 1 (Paper Writing):** According to your suggestions, we will polish the revised manuscript to enhance the readability.
>
> **Weaknesses 2 (More Discussions on the Novelty of this Submission):** Thanks for your recognition of our work. Following your suggestions, we will enhance the revised manuscript by including more discussion on our innovative contributions, such as the limitations of the proposed loss function and future research directions (as addressed in our responses to your raised **Weaknesses 4 and Limitation**).
>
> **Weaknesses 3 (More Experimental Results, e.g., Combination with More Methods):** Following your suggestion, we have supplemented more experimental results: in response to the raised **Weakness 1 by #Reviewer 1**, we include runtime evaluations for computing the proposed loss function and optimizing its corresponding objective; to address the raised **Weakness 3 by #Reviewer 1**, we perform ablation studies by individually using JMMD and HSIC; in response to the raised **Weakness 2 by #Reviewer 2**,  we conduct ablation studies by using different JMMD kernels for the proposed JMMD-HSIC. Furthermore, in **Section 6 of the supplementary materials**, we provide: (1) robustness experiments of the proposed method with different classifiers (Table 1 in Lines 117-118), (2) comparative experimental results on the Office10-Caltech10 dataset (Table 2 in Lines 117-118), and (3) comparative experimental results on the ImageCLEF-DA dataset (Table 3 in Lines 135-136).
>
> Moreover, we have incorporated the proposed loss function into both a shallow subspace-learning-based domain adaptation method (CRLP-DA21 [3]) and three end-to-end deep adaptation frameworks (JAN [1], DSAN [2], and DLP-NNM [4]), as shown in the following two tables. Although the primary objective of this work is to address the challenges of applying JMMD in shallow subspace-learning methods, our final loss function demonstrates successful integration with end-to-end deep adaptation approaches. Notably, JAN is the pioneering work published in ICML'17 that first apply JMMD into end-to-end deep adaptation frameworks, but their original paper identified a critical issue. The following sentence is originated from the ICML paper: **· · · while in conventional shallow domain adaptation methods the joint distributions are not easy to manipulate and match.** This issue serves as a key motivation for our work. DSAN represents another significant contribution that specifically addresses how to define class conditional distribution discrepancy in deep adaptation frameworks. CRLP-DA21 and DLP-NNM offer representative solutions that mitigate the impact of pseudo-labeling issues on loss functions requiring target domain pseudo-labels through innovative classifier designs. Our experimental results show that the proposed loss function achieves varying degrees of performance improvement when applied to all four of these methods, which can demonstrate the effectiveness of our proposed loss function.
>
> | 6 tasks of the Office-31 dataset | Venue |  Accuracy (Average (%)) |
> | -------------|:------------- |:-------------:|
> | JAN | ICML'17 | 84.3 |
> | JAN+JMMD-HSIC | - | **86.2** |
> | DSAN | TNNLS'20 | 88.4 |
> | DSAN+JMMD-HSIC | - | **89.8** |
> |  CRLP-DA21 | TCSVT'21 | 90.4 |
> |  CRLP-DA21+JMMD-HSIC | - | **91.3** |
> | DLP-NNM | TIP'25 | 90.9 |
> | DLP-NNM+JMMD-HSIC | - | **91.4** |
>
> | 12 tasks of the Office-Home dataset | Venue |  Accuracy (Average (%)) |
> | -------------|:------------- |:-------------:|
> | JAN | ICML'17 | 58.3 |
> | JAN+JMMD-HSIC | - | **60.5** |
> | DSAN | TNNLS'20 | 67.6 |
> | DSAN+JMMD-HSIC | - | **69.6** |
> | CRLP-DA21 | TCSVT'21 | 71.4 |
> | CRLP-DA21+JMMD-HSCI | - | **72.9** |
> | DLP-NNM | TIP'25 | 73.7 |
> | DLP-NNM+JMMD-HSCI | - | **74.3** |
>
> **Weaknesses 4 and Limitation (Limitations of the Proposed Method and Future Work):** The limitations of our proposed loss function are as follows: **(a) Domain Adaptation Setting.** The proposed JMMD-HSIC can only address the closed-set domain adaptation setting, where the label spaces of the source domain and target domain are identical. However, we do not explore its application in more complex domain adaptation scenarios, such as open-set domain adaptation [5], partial domain adaptation [6], weakly-supervised domain adaptation [7], etc. **(b) Task.** This submission only investigates the application of JMMD-HSCI in cross-domain classification tasks, while its potential in other more complex tasks, such as semantic segmentation and object detection, remains under-explored. **(c) Classifier.** Although this study has verified that JMMD can improve performance across different classifiers, how to design JMMD kernels for different classifiers has not been sufficiently explored. For future work, we plan to explore: **(1)** designing an advanced label kernel to handle more diverse domain adaptation scenarios; **(2)** extending the framework to more complex visual tasks; **(3)** designing novel label kernels by systematically analyzing how different classifiers vary in their sensitivity to feature distribution alignment and feature discriminability.
>
> [1] Long M.S., et al. Deep Transfer Learning with Joint Adaptation Networks. ICML, 2017.
>
> [2] Zhu Y. C., et al. Deep Subdomain Adaptation Network for Image Classification. TNNLS, 2020.
>
> [3] Wang W. et al. Confidence Regularized Label Propagation Based Domain Adaptation. TCSVT, 2021.
>
> [4] Wang W., et al. Deep Label Propagation with Nuclear Norm Maximization for Visual Domain Adaptation. TIP, 2025.
>
> [5] Saito K., et al. Open Set Domain Adaptation by Backpropagation. ECCV, 2018.
>
> [6] Cao Z. J., et al. Partial Adversarial Domain Adaptation. ECCV, 2018.
>
> [7] Shu Y., etal. Transferable Curriculum for Weakly-Supervised Domain Adaptation. AAAI, 2019.

---

### Official Review · Reviewer_Cj1Z · 2025-06-30

**Clarity:** 4
**Significance:** 3
**Originality:** 3
**Rating:** 5
**Confidence:** 5

**Summary:**

The distribution-distance metric has always been a critical issue in domain adaptation. In this paper, the authors first address the challenge of applying JMMD in subspace learning frameworks through a series of theoretical derivations. Based on this result, the authors uncover the uniformity of JMMD. Furthermore, from the perspective of graph embedding, the authors compare the relationship between HSIC and JMMD and explain why feature discrimination is compromised during the process of minimizing JMMD. Finally, the authors propose a novel loss function (JMMD-HSIC) and conduct extensive experiments to validate the correctness of the theoretical results and the effectiveness of the proposed loss function.

**Questions:**

Please refer to the weaknesses.

**Ethical Concerns:**

["NO or VERY MINOR ethics concerns only"]

**Final Justification:**

The authors have fully addressed all my concerns. I have also reviewed the comments from other reviewers, particularly #Reviewer 5L6J and #Reviewer J6vh, who both highlighted the significant strengths of this submission. Therefore, I maintain my recommendation to accept this submission.

**Limitations:**

Yes

**Paper Formatting Concerns:**

This paper follows the formatting instructions.

**Quality:**

4

**Strengths And Weaknesses:**

Strengths:
1. The paper is highly readable and easy to understand.
2. The supplementary materials provide thorough theoretical derivations, making both the theoretical findings and experimental results highly credible.
3. The distribution-distance metric is a crucial issue in domain adaptation, and the theoretical results in this paper not only resolve the difficulty of applying JMMD in subspace learning frameworks but also reveal the uniformity of JMMD and the reason for its degradation of discriminability. These contributions demonstrate strong originality and significantly advance the field of domain adaptation.
4. The experimental results are comprehensive, effectively validating the correctness of the theoretical results and the effectiveness of the proposed method.

Weaknesses:
1. The authors should include more discussion to explore how the theoretical findings could inspire future work.
2. Figure 1 should provide more detailed descriptions to clarify the motivation of this study and the workflow of the proposed method.
3. In domain adaptation, the labels of the target domain are unavailable, whereas JMMD requires labels from both domains. How does this paper address this issue? If pseudo-labels are adopted, how is the correct modeling of this loss ensured?
4. Can the proposed loss function be effectively applied in deep learning frameworks?

---

> ### Author Rebuttal · Authors · 2025-07-30
>
> **Weaknesses 1 (Future Work Inspired by the Theoretical Findings in this Submission):** In our responses to **#Reviewers 1, 2, and 4**, we have thoroughly discussed three key limitations of the proposed loss function. To address these limitations in future work, we plan to: **(a)** designing an advanced label kernel to handle more diverse domain adaptation scenarios; **(b)** extending the framework to more complex visual tasks; **(c)** designing novel label kernels by systematically analyzing how different classifiers vary in their sensitivity to feature distribution alignment and feature discriminability.
>
> **Weaknesses 2 (Detailed Descriptions for the Motivation and Workflow of the Proposed Method):** In our responses to **#Reviewers 1 and 2,** we clarify that the motivation of this work originates from an ICML'17 paper, where the authors argued that JMMD is difficult to apply to shallow domain adaptation methods. The following sentence is originated from the ICML paper: **· · · while in conventional shallow domain adaptation methods the joint distributions are not easy to manipulate and match.**
>
> We will provide the following detailed descriptions for Figure 1 (Lines 36-37) in our revised manuscript: The overview of our revealed theoretical results and proposed JMMD-HSIC. (a) We map the features (upper part) and labels (lower part) of the source and target domains into the RKHS, respectively; (b) The application of JMMD (upper part) and HSIC (lower part) in a subspace-learning framework; (c) The graph embedding interpretation of JMMD (upper part) and HSIC (lower part) in a subspace-learning framework; (d) Learned feature representations of the source and target domains after subspace learning. '$−$' in the module of JMMD means the feature-label dependence difference, and '$+$' in the module of HSIC means separately considering feature-label dependence in the two domains. $\textbf{X}$ and $\textbf{Y}$ are data feature and label matrices. $\psi$ and $\phi$ are feature and label mappings for a RKHS. $\textbf{T}$ and $\textbf{B}$ are projection matrices for a projected RKHS. $\textbf{T}\psi(\textbf{X})\otimes \phi (\textbf{Y})$ is a tensor-product operator for the covariance. $\text{tr}(\textbf{B}^{\top}\textbf{K}^{\textbf{X}\textbf{X}}(\textbf{K}^{\textbf{Y}\textbf{Y}}\odot\textbf{M}^{\text{J}/\text{H}})\textbf{K}^{\textbf{X}\textbf{X}}\textbf{B})$ is the deduced concise JMMD/HSIC in a projected RKHS.
>
> **Weaknesses 3 (Pseudo-Labels Issue for the Proposed Loss Function):** **(a)** As you rightly pointed out, our model employs pseudo-labeling (prediction results) and trains the model by iteratively updating both the pseudo-labels and our designed loss function. To address the inherent unreliability of pseudo-labels, this submission adopts classifiers specifically designed by two subspace-learning-based domain adaptation techniques (SPL [45], OGL2P [46]). **(b) In Section 6 of the supplementary material (Lines 116-127)**, ablation experiments comparing JMMD, HSCI and our JMMD-HSCI are performed on Office10-Caltech10 dataset, with multiple classifiers verifying the loss's robustness. Furthermore, we compare the performance of two distinct label formats (one-hot hard labels and probabilistic soft labels). As referenced in [4], this challenge can be mitigated by designing a confidence-regularized classifier that alleviates both the over-confidence problem of hard labels and the excessive noise introduced by probabilistic soft labels. Ultimately, this approach enhances the model's robustness against pseudo-label unreliability. **(c)** In the following two tables, we have also applied the proposed loss function to the shallow subspace learning-based domain adaptation framework (CRLP-L21) described in [4], demonstrating performance improvements compared with CRLP-L21. **(d)** For future work, we plan to explore: (1) confidence-based filtering methods to discard unreliable pseudo-labels, (2) enhancing the loss function's robustness to pseudo-label issue, and (3) designing more advanced classifiers to obtain more reliable pseudo-labels.
>
> **Weaknesses 4 (the Applications of Our Proposed Loss Function in Deep Learning Frameworks):** Since this study primarily addresses the challenge of applying JMMD into shallow subspace-learning-based domain adaptation frameworks, we initially do not implement the proposed loss function in end-to-end deep domain adaptation frameworks. Following your suggestion, we have now applied the proposed loss into three different deep learning frameworks (JAN [1], DSAN [2], DLP-NNM [3]), with experimental results on both the Office-31 and Office-Home datasets presented in the following two tables. It can be observed that the proposed loss function still leads to varying degrees of performance improvement across these three frameworks.
>
> | 6 tasks of the Office-31 dataset | Venue |  Accuracy (Average (%)) |
> | -------------|:------------- |:-------------:|
> | JAN | ICML'17 | 84.3 |
> | JAN+JMMD-HSIC | - | **86.2** |
> | DSAN | TNNLS'20 | 88.4 |
> | DSAN+JMMD-HSIC | - | **89.8** |
> |  CRLP-DA21 | TCSVT'21 | 90.4 |
> |  CRLP-DA21+JMMD-HSIC | - | **91.3** |
> | DLP-NNM | TIP'25 | 90.9 |
> | DLP-NNM+JMMD-HSIC | - | **91.4** |
>
> | 12 tasks of the Office-Home dataset | Venue |  Accuracy (Average (%)) |
> | -------------|:------------- |:-------------:|
> | JAN | ICML'17 | 58.3 |
> | JAN+JMMD-HSIC | - | **60.5** |
> | DSAN | TNNLS'20 | 67.6 |
> | DSAN+JMMD-HSIC | - | **69.6** |
> | CRLP-DA21 | TCSVT'21 | 71.4 |
> | CRLP-DA21+JMMD-HSCI | - | **72.9** |
> | DLP-NNM | TIP'25 | 73.7 |
> | DLP-NNM | - | **74.3** |
>
> [1] Long M.S., et al. Deep Transfer Learning with Joint Adaptation Networks. ICML, 2017.
>
> [2] Zhu Y. C., et al. Deep Subdomain Adaptation Network for Image Classification. TNNLS, 2020.
>
> [3] Wang W., et al. Deep Label Propagation with Nuclear Norm Maximization for Visual Domain Adaptation. TIP, 2025.
>
> [4] Wang W. et al. Confidence Regularized Label Propagation Based Domain Adaptation. TCSVT, 2021.

---

### Official Review · Reviewer_NVEg · 2025-07-03

**Clarity:** 2
**Significance:** 2
**Originality:** 2
**Rating:** 4
**Confidence:** 3

**Summary:**

The paper exploits the JMMD for UDA problem. To address the partial derivative problem caused by the tensor-product operator, the paper deduces a concise JMMD based on the Representer theorem. It analyzes the connection between marginal, conditional, and weighted conditional MMD. Also, the paper provides a graph embedding viewpoint for JMMD with HSIC.

**Questions:**

For the statement “hard to be applied into a subspace learning framework”, what is subspace learning? There is no clear definition in the paper.

“as its empirical estimation involves a tensor-product operator whose partial derivative is difficult to obtain.” In the modern deep learning framework, like PyTorch, can the auto-differentiation tools solve this problem?


In Figures 2c and d, why are the features of different classes overlapped?

**Ethical Concerns:**

["NO or VERY MINOR ethics concerns only"]

**Final Justification:**

Most of my concerns are addressed. So, I raise my score accordingly.

**Limitations:**

The paper does not discuss the limitations of the proposed method.

**Quality:**

2

**Strengths And Weaknesses:**

Strengths: The experimental results show the benefit of JMD-HSIC. Feature visualization of SPL+JMMD-HSIC is interesting.

Weaknesses:

Writing is hard to follow.

1. Using the Representer theorem to formulate the kernel of JMMD is not new, e.g., [1].

[1]. Long M, Wang J, Ding G, et al. Transfer feature learning with joint distribution adaptation[C]//Proceedings of the IEEE international conference on computer vision. 2013: 2200-2207.

2. Regarding Theorem 1, why is the uniformity of JMMD important for UDA? In UDA, essentially, you want to align the joint distribution $p(x,y)$. Using different characteristic kernels can reveal marginal, class-conditional, and weighted class conditional distributions from a joint distribution. Many previous MMD-based methods already attempt to align the joint distribution rather than the pure marginal or conditional distribution. Can you use the combination of the three? An ablation study on different chosen kernels would be good.


3. In lines 227-229, “From (7), we observe that JMMD will push two data points from the same classes in the same domain further, and draw those from the same classes in different domains closer. HSIC will draw two data points from the same classes in the same domain closer (intra-class compactness).” Can you elaborate on the details? It is not easy to derive such a conclusion.

4. The graph embedding viewpoint has been discussed in [2]. Can you discuss the connection of the two? Specifically, [2] has the same conclusion and a more detailed explanation for $w_{i,j}^J$.

[2]. Chen Y, Song S, Li S, et al. A graph embedding framework for maximum mean discrepancy-based domain adaptation algorithms[J]. IEEE Transactions on Image Processing, 2019, 29: 199-213.

---

> ### Author Rebuttal · Authors · 2025-07-30
>
> **Weaknesses 1 (the Novelty of JMMD's Kernel Formulated by the Representer Theorem):** **First,** The motivation of this submission mainly comes from a paper published at ICML’17 [3] (precisely the authors of the reference you mentioned [1]), and they applied JMMD into a deep neural network. They argued that JMMD is difficult to be applied into shallow domain adaptation methods. The following sentence is originated from the ICML paper: **“· · · while in conventional shallow domain adaptation methods the joint distributions are not easy to manipulate and match.”** **Second,** JDA (Joint Distribution Adaptation) proposed in the reference you mentioned [1] defines the joint distribution difference as the combination of the marginal and class conditional MMD (i.e., their summation), which differs from our definition of joint distribution difference. From the uniformity of JMMD proved in our work, JDA actually represents a special case of JMMD. Consequently, **the kernel obtained by JDA through the Representer theorem corresponds to the sum of the marginal kernel and class conditional kernel, rather than the JMMD kernel.** **Third,** their definitions of marginal and class conditional MMD do not involve complex tensor product operations, making their problem inherently less challenging than ours. **Fourth,** when employing the Representer theorem to derive the JMMD kernel (thereby eliminating its complicated tensor-product operation), we not only uncover its uniformity but also explain—from a graph embedding perspective—the fundamental reason why it compromises feature discriminability. This insight could lead us to propose a novel and more robust loss function, such as the proposed JMMD-HSIC.
>
> **Weaknesses 2 (the Importance of JMMD's Uniformity for UDA):** **First,** as addressed in our reply to your raised **Weakness 1**, our definition of joint distribution difference differs from theirs. By proving the uniformity of JMMD, we identify their method as a special case of ours. This does not imply equivalence, since we can improve JMMD by exploring its limitations and defining more robust kernels. For instance, this submission reveals how JMMD compromises feature discriminability and accordingly designs the JMMD-HSIC kernel. **Second,** our approach combines not the three components but the class conditional distribution kernel and our newly designed HSIC-based kernel. The rationale for selecting the class conditional distribution kernel is twofold: **(a)** Marginal distribution kernel neglect label information, often leading to imprecise distribution alignment; **(b)** While weighted class-conditional kernel can handle class imbalance issue, our experimental datasets exhibit minimal class imbalance, making their results nearly indistinguishable from the class conditional kernel. **Third,** following your suggestion, we have validated these claims through ablation studies, as shown in the following two tables. **Fourth,** the uniformity of JMMD enables our to develop more robust loss functions (as demonstrated by our JMMD-HSIC) by identifying its limitations.
>
> | 6 tasks of the Office-31 dataset | Accuracy (Average (%)) |
> | ------------- |:-------------:|
> | SPL | 89.6 |
> | SPL+JMMD-HSIC (marginal kernel) | 85.6 |
> | SPL+JMMD-HSIC (class conditional kernel) | **91.0** |
> | SPL+JMMD-HSIC (weighted class conditional kernel) | 90.9 |
> | OGL2P | 90.7 |
> | OGL2P+JMMD-HSIC (marginal kernel) | 86.7 |
> | OGL2P+JMMD-HSIC (class conditional kernel) | **91.5** |
> | OGL2P+JMMD-HSIC (weighted class conditional kernel) | **91.5** |
>
> | 12 tasks of the Office-Home dataset | Accuracy (Average (%)) |
> | ------------- |:-------------:|
> | SPL | 71.0 |
> | SPL+JMMD-HSIC (marginal kernel) | 64.9 |
> | SPL+JMMD-HSIC (class conditional kernel) | **71.9** |
> | SPL+JMMD-HSIC (weighted class conditional kernel) | 71.8 |
> | OGL2P | 72.8 |
> | OGL2P+JMMD-HSIC (marginal kernel) | 69.8 |
> | OGL2P+JMMD-HSIC (class conditional kernel) | **73.6** |
> | OGL2P+JMMD-HSIC (weighted class conditional kernel) | **73.6** |
>
> **Weaknesses 3 (Details about the Sentences in Lines 227-229):** **First,** the primary objective of graph embedding is to minimize the objective function (6) (Lines 219-220). When the weight wij between samples i and j is positive, we should reduce the feature distance (**draw them closer**) between them to prevent the objective function from increasing; conversely, when wij is negative, we should increase their feature distance (**push them further**) to decrease the objective function value. **Second,** through derivation, we obtain (8) (Lines 224-225) and find it maintains the same form as the objective function in (6). From the JMMD (side of (8)) and its corresponding weights (left side of (7) (Lines 223-224)), we observe that when samples belong to the same domain and same class, the weight is negative, indicating we should increase the distance or **push them further between two data points from the same classes in the same domain**. Conversely, when samples are from different domains but share the same class, the weight is positive, suggesting we should decrease their distance or **draw them closer between two data points from the same classes in the same domain**. Similarly, analyzing the HSIC (right side of (8)) and its weights (right side of (7)), we find positive weights occur when samples belong to the same domain and the same class, meaning we should decrease the distance or **draw them closer between two data points from the same classes in the same domain**.
>
> **Weaknesses 4 (the Connections between [2] and Ours):** The connections of the two are as follows: **Special Case Analysis:** Their work actually reveals a special case within our JMMD framework - specifically how marginal MMD and class conditional MMD may compromise discriminability. Our formulation generalizes to more distribution metrics satisfying JMMD's concise form, including the weighted class conditional MMD and the proposed JMMD-HSCI. **Different Solution Strategies:** Our approach fundamentally differs in addressing discriminability degradation. We introduce the JMMD-HSIC kernel with proven its reproducibility, maintaining its validity as a special case of JMMD. Their approach lacks strong theoretical foundation as it modifies the marginal MMD and class conditional MMD empirically through observation.
>
> **Question 1 (the Definition of Subspace Learning in this Submission):** In response to your raised **Weakness 2**, we clarify that the motivation of this work originates from an ICML'17 paper, where the authors argued that JMMD is difficult to apply to shallow domain adaptation methods. The following sentence is originated from the ICML paper: **· · · while in conventional shallow domain adaptation methods the joint distributions are not easy to manipulate and match.** Therefore, the subspace-learning methods referred to in this submission specifically denote shallow subspace-learning frameworks such as Principal Component Analysis (PCA) and Non-negative Matrix Factorization (NMF), etc.
>
> **Question 2 (Hard to Obtain Partial Derivative):** In response to your raised **Question 1**, we have provided clarification regarding the subspace-learning framework mentioned in this submission. The primary objective of this work is to address the challenge of obtaining partial derivatives when applying JMMD into the shallow subspace-learning frameworks **not the deep subspace-learning frameworks**.
>
> **Question 3 (Classes Overlapped in Figure 2c and d (Lines 300-301)):** Under the domain adaptation setting where target domain labels are unavailable during training, we follow the common practice in the field by employing pseudo-labels (model predictions) to iteratively update the pseudo-labels and our proposed loss function. However, for the visualization in this Figure, we adopted the ground-truth labels following Reference [1]. **The inherent unreliability of pseudo-labels remains a critical bottleneck limiting domain adaptation performance. Consequently, it is inherently challenging to learn features that can effectively separate different classes according to their true labels. The observed inter-class overlap primarily stems from the discrepancy between pseudo-labels and ground-truth labels for these samples.**
>
> **Limitation (Limitation Discussions for the Proposed Method):** The limitations of our proposed loss are as follows: **(a) Domain Adaptation Setting.** The proposed JMMD-HSIC can only address the closed-set domain adaptation setting, where the label spaces of the source domain and target domain are identical. However, we do not explore its application in more complex domain adaptation scenarios, such as open-set domain adaptation [4], partial domain adaptation [5], weakly-supervised domain adaptation [6], etc. **(b)** Task. This submission only investigates the application of JMMD-HSCI in cross-domain classification tasks, while its potential in other more complex tasks, such as semantic segmentation and object detection, remains under-explored. **(c)** Classifier. Although this study has verified that JMMD can improve performance across different classifiers, how to design JMMD kernels for different classifiers has not been sufficiently explored.
>
> [1] Long M.S., et al. Transfer Feature Learning with Joint Distribution Adaptation. ICCV, 2013.
>
> [2] Chen Y. M., et al. A Graph Embedding Framework for Maximum Mean Discrepancy-based Domain Adaptation Algorithms. TIP, 2019.
>
> [3] Long M.S., et al. Deep Transfer Learning with Joint Adaptation Networks. ICML, 2017.
>
> [4] Saito K., et al. Open Set Domain Adaptation by Backpropagation. ECCV, 2018.
>
> [5] Cao Z. J., et al. Partial Adversarial Domain Adaptation. ECCV, 2018.
>
> [6] Shu Y., etal. Transferable Curriculum for Weakly-Supervised Domain Adaptation. AAAI, 2019.

---

> > ### Author Response · Authors · 2025-08-05
> > **Many Thanks**
> >
> > Thanks for your insightful comments again. Having provided a substantive response to your feedback and emphasized our key contributions, we respectfully hope for a reconsideration of the score. Please let us know if you need additional points to discuss.
> >
> > Best regards,
> >
> > Authors of #14339.

---

> > ### Comment · Reviewer_NVEg · 2025-08-05
> >
> > Thanks for the authors' response. Most of my concerns are addressed.

---

> > > ### Author Response · Authors · 2025-08-05
> > >
> > > We appreciate your recognition of our responses and would like to thank you again for your diligent review work.
> > >
> > > Best regards,
> > >
> > > Authors of #14339.

---

### Official Review · Reviewer_5L6J · 2025-07-04

**Clarity:** 3
**Significance:** 3
**Originality:** 3
**Rating:** 5
**Confidence:** 3

**Summary:**

The paper addresses the problem of domain adaptation in which the task is to reduce the discrepancy between the joint probability distribution between the source and target domain. The paper builds upon the joint maximum mean discrepancy (JMMD) metric and discusses its limitations of being applied into a subspace-learning framework as its empirical estimation involves a tensor-product operator whose partial derivative is difficult to obtain. The authors deduce a concise JMMD based on the Representer theorem that avoids the tensor-product operator and obtain two essential findings. They reveal the uniformity of JMMD by proving that previous marginal, class conditional, and weighted class conditional probability distribution distances are three special cases of JMMD with different label reproducing kernels. Moreover the authors also discuss the relationship between JMMD and HSIC based on graph embeddings. Based on these observations, the paper proposes a novel loss JMMD-HSIC by jointly considering JMMD and HSIC to promote discrimination of JMMD. Experiments on four benchmarks are shown.

**Questions:**

All of my concerns are listed above in the weaknesses section. The paper is well written and explained. The experiments section of the paper can be made stronger.

**Ethical Concerns:**

["NO or VERY MINOR ethics concerns only"]

**Final Justification:**

The authors have addressed all of my concerns, and I am raising my score. I hope the additional information from the rebuttal is reflected in the final version of the paper.

**Limitations:**

No. The authors have mentioned that they did not discuss the limitations because of space constraints, but there is no limitation discussed in the supplementary as well.

**Quality:**

3

**Strengths And Weaknesses:**

**Strengths:**
* The paper is well written and all the concepts are well explained.
* The relationship analysis and the joint optimization of JMMD and HSIC is novel.
* The graph embedding viewpoint discussed in Section 3.4 showing findings like JMMD degrades the feature discrimination is interesting and insightful.




**Weaknesses:**
* It would be great to have convergence analysis for the loss function - including speed, time, etc.
* Robustness analysis: are the reported values averaged over multiple runs? If yes/no, what do the standard deviations look like? How robust is the loss function?
* Ablation study: Although the authors have provided some ablation studies in Figure 3, it would be great to check the values in Table 1 or 2 by individually employing the JMMD and HSIC components.
* What projection algorithm was used for visualizing the features in Figure 2.
* Typo: Figure 1 (c) - draw close and push away legends using the same symbol.

---

> ### Author Rebuttal · Authors · 2025-07-30
>
> **Weakness 1 (Convergence Analysis for the Proposed Loss Function):** **(a) Differentiable.** Since the proposed loss function is differentiable, the final objective function to be optimized is also convergent as long as the applied subspace-learning-based domain adaptation framework is convergent. **(b) Closed-Form Solution.** In this manuscript, the proposed loss function is applied into JDA [15], SPL [45], and OGL2P [46], and the resulting objective functions to be optimized all admit closed-form solutions, ensuring convergence. **(c) Computational Complexity.** The computational complexity of the proposed loss function is $O(\text{Cn + Cn}^\text{2})$, where $\text{C}$ is the number of classes and $\text{n}$ is the total number of samples from both the source and target domains. Since $\text{C}$ is generally much smaller than the sample size $\text{n}$, the complexity mainly depends on the number of samples. **(d) Time/Speed.** We analyze the time required for the algorithm to compute the proposed loss function (**loss construction**) and optimize the corresponding objective function (**objective optimization**) on two datasets and two subspace-learning-based domain adaptation frameworks (SPL and OGLP2P), with the results recorded in the following two tables.
>
> **(a)** It can be observed that due to the use of MATLAB, which provides highly efficient matrix computation capabilities, the objective optimization and the loss construction exhibit highly efficient runtime performance. **(b)** Since the Office-Home dataset is larger in scale, it requires more time compared to the Office-31 dataset. **(c)** the subspace-learning-based domain adaptation framework applied in this manuscript belongs to a two-stage domain adaptation approach. That is, deep learning is first employed to extract a promising feature representation, followed by further refinement using subspace-learning methods. Since subspace-learning methods entail significantly lower computational overhead compared to deep learning training, they hold substantial research value and have consistently garnered sustained attention from researchers. **(d)** These results not only validate the efficiency of the two-stage domain adaptation approach but also highlight the necessity of addressing the challenges in applying JMMD within subspace-learning frameworks.
>
> | 6 tasks of the Office-31 dataset | Time/Speed (Average (seconds)) |
> | ------------- |:-------------:|
> | SPL (objective optimization)      | 0.357 |
> | SPL (loss construction)     | 0.075 |
> | OGL2P (objective optimization) | 1.330 |
> | OGL2P (loss construction) | 0.085 |
>
> | 12 tasks of the Office-Home dataset | Time/Speed (Average (seconds)) |
> | ------------- |:-------------:|
> | SPL (objective optimization)      | 1.253 |
> | SPL (loss construction)     | 0.303 |
> | OGL2P (objective optimization) | 0.557 |
> | OGL2P (loss construction) | 0.310 |
>
> **Weakness 2 (Robustness Analysis for the Proposed Loss Function):** The results reported in this manuscript **are not averaged over multiple runs**, for the following reasons: The proposed loss function is applied into three shallow subspace-learning-based domain adaptation methods. Since no stochastic processes are involved, the results obtained from each run are deterministic, with a standard deviation of 0. Referring to the original experimental results in these three papers (JDA [15], SPL [45], OGLP [46]), none of them record or mention the standard deviation.
>
> The loss function we designed is robust, free from stochastic processes, and ensures that the same input will always produce the same output. **In Section 6 of the supplementary material (Lines 116-127) and Figure 3 of the main text (Lines 300-301)**, we conduct ablation studies on Office10-Caltech10 dataset comparing JMMD, HSCI, and our proposed JMMD-HSCI. Different classifiers are employed to validate the robustness of the proposed loss function. Additionally, in our response to your raised **Weakness 6**, we have provided a detailed discussion regarding the limitations of the proposed loss function.
>
> **Weakness 3 (Ablation Study by Individually Employing the JMMD and HSIC Components):** **In Section 6 of the supplementary material (Lines 116-127)**, **ablation experiments comparing JMMD, HSCI and our JMMD-HSCI** are performed on Office10-Caltech10 dataset, with multiple classifiers verifying the loss's robustness. As per your suggestions, we present the experimental results individually using JMMD and HSIC in the following two tables. **(a)** Considering that the classifiers employed in SPL and OGL2P frameworks require higher feature discriminability, we can observe that HSIC outperforms JMMD. **(b)** Since the Office-31 dataset has smaller distribution discrepancies, the performance improvement of HSIC over JMMD is more significant on the Office-31 dataset (2.6%, 2.1%) compared to the Office-Home dataset (0.3%, 0.2%). **(c)** Our proposed loss function achieves optimal results. Although both SPL and OGL2P methods also incorporate loss functions for feature distribution alignment and discriminative feature learning, our approach provides more precise balancing between these two objectives by fundamentally analyzing how JMMD compromises feature discriminability. In other words, we more effectively mitigate the impact of distribution alignment on feature discriminability. Consequently, our method achieves superior performance compared to SPL and OGL2P.
>
> | 6 tasks of the Office-31 dataset | Accuracy (Average (%)) |
> | ------------- |:-------------:|
> | SPL | 89.6 |
> | SPL+JMMD | 87.3 |
> | SPL+HSIC | 89.9 |
> | SPL+JMMD-HSIC | **91.0** |
> | OGL2P | 90.7 |
> | OGL2P+JMMD | 88.5 |
> | OGL2P+HSIC | 90.6 |
> | OGL2P+JMMD-HSIC | **91.5** |
>
> | 12 tasks of the Office-Home dataset | Accuracy (Average (%)) |
> | ------------- |:-------------:|
> | SPL | 71.0 |
> | SPL+JMMD | 68.7 |
> | SPL+HSIC | 69.0 |
> | SPL+JMMD-HSIC | **71.9** |
> | OGL2P | 72.8 |
> | OGL2P+JMMD | 71.3 |
> | OGL2P+HSIC | 71.5 |
> | OGL2P+JMMD-HSIC | **73.6** |
>
> **Weakness 4 (Projection Algorithm for Feature Visualization in Figure 2 (Lines 300-301)):** In Figure 2, we employ the **t-SNE** algorithm for feature visualization, which has been widely adopted in domain adaptation field [45,46].
>
> **Weakness 5 (Typo in Figure 1 (c) (Lines 36-37)):** Thank you for identifying this typo. We will correct it in the revised manuscript.
>
> **Limitation (Limitation Discussions for the Proposed Loss Function):** The limitations of our proposed loss are as follows: **(a) Domain Adaptation Setting.** The proposed JMMD-HSIC can only address the closed-set domain adaptation setting, where the label spaces of the source domain and target domain are identical. However, we do not explore its application in more complex domain adaptation scenarios, such as open-set domain adaptation [1], partial domain adaptation [2], weakly-supervised domain adaptation [3], etc. **(b) Task.** This submission only investigates the application of JMMD-HSCI in cross-domain classification tasks, while its potential in other more complex tasks, such as semantic segmentation and object detection, remains under-explored. **(c) Classifier.** Although this study has verified that JMMD can improve performance across different classifiers, how to design JMMD kernels for different classifiers has not been sufficiently explored.
>
> [1] Saito K., et al. Open Set Domain Adaptation by Backpropagation. ECCV, 2018.
>
> [2] Cao Z. J., et al. Partial Adversarial Domain Adaptation. ECCV, 2018.
>
> [3] Shu Y., etal. Transferable Curriculum for Weakly-Supervised Domain Adaptation. AAAI, 2019.

---

> > ### Author Response · Authors · 2025-08-05
> > **Many Thanks**
> >
> > Thank you once again for your valuable guidance. We have thoroughly addressed your feedback. Should you have any additional questions, please do not hesitate to let us know.
> >
> > Best regards,
> >
> > Authors of #14339.

---

> > ### Comment · Reviewer_5L6J · 2025-08-06
> > **Reply to the rebuttal**
> >
> > Thanks for the rebuttal and for addressing all of my concerns. I hope the additional information from the rebuttal is reflected in the final version of the paper.

---

> > > ### Author Response · Authors · 2025-08-09
> > > **Many Thanks**
> > >
> > > Thank you for your recognition of our response to your comments, as well as your ongoing and diligent review of our submission.
> > >
> > > Best regards,
> > >
> > > Authors of #14339.

---

### Decision · Program_Chairs · 2025-09-17

**Decision:**

Accept (oral)

**Comment:**

This paper was reviewed by 4 experts in the field and received three "Accept" and one "Borderline Accept" as the final ratings. The reviewers agreed that the joint optimization of joint maximum mean discrepancy (JMMD) and Hilbert Schmidt independence criterion (HSIC) is a novel idea, the paper is well-written with sound theoretical derivations, the experimental analysis is comprehensive, and the results are promising.

Reviewer 5L6J mentioned a point about the convergence analysis of the proposed loss function, which was addressed convincingly by the authors in the rebuttal. The authors also conducted experiments on the Office-31 and Office-Home datasets to analyze the convergence speed of their loss function. Concerns were also raised regarding the individual contributions of the JMMD and HSIC components; the authors have conducted ablation studies on the Office-31 and Office-Home datasets and have demonstrated the results with each component separately, and the benefits of combining them. The authors have also conducted experiments to show that their proposed loss function can be applied to train deep neural networks, in response to a comment from Reviewer Cj1Z.

The reviewers, in general, have a positive opinion about the paper and its contributions; during the post-rebuttal discussion period, all of them have mentioned that their concerns were appropriately addressed. Based on the reviewers' feedback, the decision is to recommend the paper for acceptance to NeurIPS 2025. The reviewers have provided some valuable comments, such as discussions about the limitations of the proposed method, and how the theoretical findings could inspire future work. The authors are encouraged to include these in the final version of their paper. Given the practical usefulness of domain adaptation, the novelty of the proposed solution, the sound theoretical analysis, the comprehensive experiments, the promising results, and the consistent positive feedback from the reviewers, the AC feels this is a top quality paper, and recommends an oral presentation at NeurIPS 2025. We congratulate the authors on the acceptance of their paper!